# Giant spin-to-charge conversion at an all-epitaxial single-crystal-oxide Rashba interface with a strongly correlated metal interlayer

Shingo Kaneta-Takada [1] ✉, Miho Kitamura[2], Shoma Arai[1], Takuma Arai[1], Ryo Okano[1], Le Duc Anh [1,3], Tatsuro Endo[1], Koji Horiba[2], Hiroshi Kumigashira [2,4], Masaki Kobayashi[1,5], Munetoshi Seki [1,5], Hitoshi Tabata [1,5], Masaaki Tanaka [1,5] ✉ & Shinobu Ohya [1,5] ✉

The two-dimensional electron gas (2DEG) formed at interfaces between $SrTiO_3$ (STO) and other oxide insulating layers is promising for use in efficient spin-charge conversion due to the large Rashba spin-orbit interaction (RSOI). However, these insulating layers on STO prevent the propagation of a spin current injected from an adjacent ferromagnetic layer. Moreover, the mechanism of the spin-current flow in these insulating layers is still unexplored. Here, using a strongly correlated polar-*metal* $LaTiO_{3+\delta}$ (LTO) interlayer and the 2DEG formed at the LTO/STO interface in an all-epitaxial heterostructure, we demonstrate giant spin-to-charge current conversion efficiencies, up to ~190 nm, using spin-pumping ferromagnetic-resonance voltage measurements. This value is the highest among those reported for all materials, including spin Hall systems. Our results suggest that the strong on-site Coulomb repulsion in LTO and the giant RSOI of LTO/STO may be the key to efficient spin-charge conversion with suppressed spin-flip scattering. Our findings highlight the hidden inherent possibilities of oxide interfaces for spin-orbitronics applications.

The interconversion phenomena between charge and spin currents at material interfaces have attracted much attention because of their potential application in the highly efficient control of magnetization in next-generation spin-orbitronics devices[1]. The spin-to-charge conversion at interfaces, the so-called inverse Edelstein effect (IEE), mainly originates from the Rashba spin-orbit interaction (RSOI) induced by the broken spatial inversion symmetry of interfaces[2]. The IEE has the great advantage of being artificially designed and controlled by creating interfaces that combine various materials[2–9]. At material interfaces, the RSOI resolves the spin degeneracy of the electronic band, splitting it into up- and down-spin bands. In the simple single parabolic band picture shown in Fig. 1a, the Fermi surface splits into large and small circles. When a spin current is injected into these bands, the chemical potential of up-spin electrons increases, while that of down-spin electrons decreases (top of Fig. 1a). Thus, the large and small Fermi surfaces shift in opposite directions, and electrons move

[1]Department of Electrical Engineering and Information Systems, The University of Tokyo, 7-3-1 Hongo, Bunkyo-ku, Tokyo 113-8656, Japan. [2]Photon Factory, Institute of Materials Structure Science, High Energy Accelerator Research Organization (KEK), 1-1 Oho, Tsukuba, Ibaraki 305-0801, Japan. [3]PRESTO, Japan Science and Technology Agency, 4-1-8 Honcho, Kawaguchi, Saitama 332-0012, Japan. [4]Institute of Multidisciplinary Research for Advanced Materials (IMRAM), Tohoku University, Sendai, Miyagi 980–8577, Japan. [5]Center for Spintronics Research Network (CSRN), The University of Tokyo, 7-3-1 Hongo, Bunkyo-ku, Tokyo 113-8656, Japan. ✉e-mail: skaneta@cryst.t.u-tokyo.ac.jp; masaaki@ee.t.u-tokyo.ac.jp; ohya@cryst.t.u-toky.ac.jp

**Fig. 1 | Sample structure and characterizations. a** Schematic illustration of the inverse Edelstein effect in a single parabolic band model. RSOI resolves the spin degeneracy of the electronic band and splits it into up- and down-spin bands (orange and blue curves). The Fermi surface splits into large and small circles with opposite spin chirality (arrows). When a spin current is injected into these bands, the chemical potential of the up-spin electrons increases (filled orange circles), while that of the down-spin electrons decreases (open blue circles) (top figure of **a**). Thus, the large and small Fermi surfaces shift in opposite directions. Due to the difference in the size of the circles, the movement of the outer circle is dominant, and in total, the electrons move toward the right side (bottom of **a**), resulting in spin-to-charge conversion. **b** Schematic structure of the LSMO/LTO/STO sample used for spin pumping measurements. **c** RHEED oscillation obtained by monitoring the (10) spot during the MBE growth of LTO. The orange and green regions are the time periods during which the shutters of the La and Ti cells are open, respectively. **d** In situ RHEED patterns of the STO substrate, LTO (3 u.c.) and LSMO (30 u.c.) taken along the [100] direction of the substrate. **e** HAADF-STEM image of LSMO (30 u.c.)/ LTO (3 u.c.)/STO.

to the right overall (bottom of Fig. 1a), resulting in a spin-to-charge current conversion. The IEE has been observed in topological insulators[8] and heterostructures of heavy metals[2,9] and oxides[3–6]. Among these, the two-dimensional electron gas (2DEG) formed at interfaces between perovskite-oxide SrTiO₃ (STO) and other oxides, such as LaAlO₃ (LAO)[10] and AlOₓ[11], is capable of efficient spin-to-charge current conversion[3–7]. Thus far, large values of conversion efficiency of the IEE, called the inverse Edelstein length $\lambda_{IEE}$, of up to ~60 nm have been reported for AlOₓ/STO[6]. This notable characteristic has been attributed to the complicated quantized multiorbital band structure with a topological feature at the STO interfaces[5,12,13]. However, both LAO and AlOₓ are insulators, which in principle hamper the direct transport of the spin current[14–16], preventing the use of the full potential of 2DEG at the STO interface.

The 2DEG formed at the polar-metal LaTiO$_{3+\delta}$ (LTO)/nonpolar-insulator STO interface is an alternative candidate[17–23]. Although LTO is an antiferromagnetic strongly correlated Mott insulator with a Ti³⁺ (3$d^1$) state in the native state, it usually transitions to a paramagnetic metal due to slight excess oxygen[19,20] or lattice distortion[21]. This metallic nature of LTO is desirable for the efficient transport of the spin current. Furthermore, unlike ordinary metals, in which itinerant *s,p* carriers dominate transport, LTO has only up-spin electrons of the relatively localized *d* orbital on a Ti site at the Fermi level. The down-spin states exist far above the Fermi level due to the strong on-site Coulomb repulsion of the strongly correlated system. Therefore, one may expect significant suppression of spin scattering during spin-current transport (see Supplementary Note 1). Another prominent feature of the LTO/STO interface is the giant Rashba coefficient $\alpha_R$, up to $1.8 \times 10^{-11}$ eVm[18], which is an order of magnitude larger than that reported for LAO/STO ($3.4 \times 10^{-12}$ eVm)[24] owing to the strong polar interface[18] and small work function of 2.59 eV for LTO (c.f. 3.25 eV for LAO)[25]. In a simple picture, $\lambda_{IEE}$ is proportional to the Rashba coefficient $\alpha_R$ (=$\lambda_{IEE}\hbar/\tau_e$), which is proportional to the energy splitting between up- and down-spin bands at a given wavenumber, where $\hbar$ is the Dirac constant and $\tau_e$ is the momentum/spin relaxation time of electrons. Therefore, the LTO/STO interface is expected to be an ideal stage for efficient spin-charge conversion. We can incorporate a coherently grown single-crystal LTO layer into perovskite-oxide heterostructures due to their excellent lattice matching[20]. However, growing high-quality LTO is generally difficult because LTO easily changes to the La₂Ti₂O₇ phase[19,20]. Here, by carefully growing a high-quality all-epitaxial La₀.₆₇Sr₀.₃₃MnO₃ (LSMO)/LTO/STO heterostructure by molecular beam epitaxy (MBE), we demonstrate giant $\lambda_{IEE}$ values, up to ~190 nm, at the LTO/STO interface using spin pumping experiments.

## Results
### Sample growth and characterizations
The sample used for our experiments is a single-crystal heterostructure of LSMO [30 unit cells (u.c.) ≈12 nm]/LTO (3 u.c. ≈1.2 nm) grown on a TiO₂-terminated STO (001) substrate by MBE (Fig. 1b). We set the LTO thickness to 3 u.c. because 2DEG transport becomes dominant over the conduction of the LTO layer at this thickness[18]. Regarding growth, we control each layer thickness precisely by monitoring the oscillation of in situ reflection high-energy electron diffraction (RHEED) with a shuttered growth technique[26] (Fig. 1c, d) (see Methods). Since LTO easily transforms into La₂Ti₂O₇ when there is excess oxygen, we grow LTO with low oxygen pressure at ~10⁻⁷ Pa[20,23]. Meanwhile, the growth of LSMO requires a much higher oxygen flux. To prevent phase transformation of LTO, we divide the growth process of LSMO into three steps with different oxygen (ozone) pressures from low (10⁻⁷ Pa) to high (10⁻⁴ Pa) (see Methods). We note that LTO grown on STO is overoxidized due to the easy diffusion of oxygen atoms from STO to LTO[20,23,27,28]. High-angle annular dark-field scanning transmission electron microscopy (HAADF-STEM) and crystallographic/

morphological analyses show that the LSMO/LTO/STO sample has a high-quality single crystallinity and abrupt interfaces without a discernible $La_2Ti_2O_7$ phase (see Supplementary Note 2 and Fig. 1e). In addition, we grow reference samples comprising LSMO (30 u.c.)/LTO (3 u.c.) on an $(LaAlO_3)_{0.3}(Sr_2TaAlO_6)_{0.7}$ (LSAT) (001) substrate and LSMO (30 u.c.) on an STO (001) substrate under the same growth conditions as those used for the LSMO/LTO/STO sample. The LSMO/LTO/LSAT sample does not have a 2DEG, so we can use it to check whether the LTO layer itself has any influence on the IEE. The LTO layer in the LSMO/LTO/STO and LSMO/LTO/LSAT samples becomes metallic due to the high oxygen pressure used for growing LSMO[20]. Using the reference LSMO/STO sample, we can eliminate the possible influence of the rectification effects of LSMO in the spin pumping measurements. We also grow a reference sample of LTO (3 u.c.)/STO for resonant angle-resolved photoemission spectroscopy (R-ARPES) measurements.

As shown in Fig. 2a, the sheet resistance $R_{sheet}$ of LSMO/LTO/STO is much lower than that of LSMO/LTO/LSAT, supporting the expected existence of the 2DEG channel only in the LSMO/LTO/STO sample. As explained later in the section of Resonant angle-resolved photoemission spectroscopy measurements and theoretical calculations, our R-ARPES measurements for the LTO (3 u.c.)/STO sample ensure the 2D feature of this electron channel; the band dispersion and the Fermi surface are reproduced well by the quantized $d$ electrons of STO, indicating the 2D confinement of the carriers[29]. We can also confirm the metallicity of LTO made with our growth conditions in Supplementary Fig. 5. As shown in Fig. 2b, the magnetic moment of the Mn atoms of LSMO/LTO/STO is the same as that of LSMO/LTO/LSAT, reflecting the same growth conditions of LSMO for both films, and both samples show high Curie temperatures above room temperature. When a magnetic field $\mu_0 H$ is applied along the [110] direction of the STO and LSAT substrates, both samples show clear hysteresis (Fig. 2c). The easy axis of magnetization is the [110] direction for both films (see ref. 4 and Supplementary Fig. 6).

**Spin pumping measurements**

To evaluate $\lambda_{IEE}$ in the 2DEG at the LTO/STO interface, we conduct spin pumping measurements using ferromagnetic resonance (FMR)[30], in which the spin current is injected from LSMO and converted into a current in the 2DEG (Fig. 3a). As shown in the $\mu_0 H$ dependence of the electromotive force (EMF) $V(H)$ measured at 15 K with $\theta_H = 0$ and 180° (Fig. 3b), $V$ is sharply enhanced at the FMR field $\mu_0 H_{FMR}$. Here, $\theta_H$ is defined as the out-of-plane angle of $H$ with respect to the in-plane [110] axis (see Fig. 1b). To extract a pure IEE signal, we decompose $V$ into a symmetric (Lorentzian) component $V_{sym}$, which includes signals of the IEE, and an antisymmetric (anti-Lorentzian) component $V_{asym}$. We see a linear relation between the microwave power $MP$ and $V_{sym}$ (inset of Fig. 3b), which is a reasonable result because the IEE signal is proportional to the amount of the spin current that has a linear relation with $MP$ (see Eq. (2) in Methods). To eliminate the contribution of the Seebeck effect, which does not depend on the sign of $H$, we use the average $V$ value $V_{sym,ave} = (V_{sym,0°} - V_{sym,180°})/2$ for the estimation of the two-dimensional current $j_{C,sym}^{2D} = V_{sym,ave}/R_{sheet}l$ for deriving $\lambda_{IEE}$, where $V_{sym,0°}$ and $V_{sym,180°}$ are the $V_{sym}$ values for $\theta_H = 0°$ and 180°, respectively[31], and $l$ is the electrode distance (1.2 mm). We note that the $V_{sym,0°}$ and $V_{sym,180°}$ data are almost symmetric as shown in Fig. 3b, indicating that the Seebeck effect is negligibly small. We estimate the spin current density $j_S^0$ from the difference in the damping constant $\alpha$ between LSMO/LTO/STO (Fig. 3a) and LSMO/STO, where we use a very low value reported for a high-quality LSMO film[32] as the damping constant of LSMO/STO (see Methods). Since the damping constant is thought to depend on the crystal quality, this choice means that we may overestimate $j_S^0$ and thus underestimate $\lambda_{IEE}$.

As shown in Fig. 3c, the two-dimensional current per microwave magnetic field square and the EMF obtained at $\theta_H = 0°$ (defined as $j_{C,0°}^{2D}/h_{rf}^2$ and $V_{0°}$, respectively) increase drastically with decreasing temperature. This behavior is completely different from the galvanomagnetic effects of LSMO, such as the planar Hall effect, which are proportional to $\rho^n$ and thus decrease with decreasing temperature[4] (see Fig. 2a). Here, $\rho$ is the resistivity of LSMO and $n$ is 1–2[33]. Therefore, the influence of the galvanomagnetic effects is negligibly small. The $j_{C,sym}^{2D}/h_{rf}^2$ value reaches 655.1 mA m$^{-1}$ G$^{-2}$ at 15 K, which is much larger than those reported previously (1.5 mA m$^{-1}$ G$^{-2}$ for Ag/Bi[2] and 53.3 mA m$^{-1}$ G$^{-2}$ for LAO/STO[3]) on the IEE, indicating that a giant spin-to-charge conversion occurs at LTO/STO. As shown in Fig. 3d, the $j_{C,0°}^{2D}/h_{rf}^2$ obtained for LSMO/LTO/LSAT (without the 2DEG) is almost zero, indicating that the spin-to-charge conversion induced at the LSMO/LTO interface and in the LTO layer is negligibly small. The $j_{C,0°}^{2D}/h_{rf}^2$ obtained for LSMO/STO is also almost zero, indicating no influence of the rectification effect of LSMO. Thus, the spin-to-charge conversion induced at the LTO/STO interface is dominant.

In Fig. 3e, we show our data for $\lambda_{IEE}$ ($= j_{C,sym}^{2D}/j_S^0$) as a function of temperature ($T$) along with the data reported for various STO interfaces with other materials[3–6,34]. The substantial increase in $\lambda_{IEE}$ with decreasing $T$ arises from the increase in the $\tau_e$ of STO due to the large increase in the dielectric constant[35] (see Supplementary Note 3), confirming that this result originates from the intrinsic IEE[4,34]. Here, we obtain a large $\lambda_{IEE}$ of 193.5 nm at 15 K, showing the superiority of the LTO/STO system; $\lambda_{IEE}$ has been reported to be 2.0 nm at HgCdTe/HgTe[8], 0.3 nm at Ag/Bi[2], 60 nm at AlO$_x$/STO[6] and 6.7 nm[4] (or −6.4 nm[3] when gated) at LAO/STO. In the spin Hall effect of bulk materials, $\lambda_{IEE}$ can be regarded as equivalent to $\theta_{SHE}\lambda_S$, where $\theta_{SHE}$ is the spin Hall angle and $\lambda_S$ is the spin diffusion length[36]. The typical values of $\theta_{SHE}\lambda_S$ for metals such as Pt and W are below 1 nm. The highest value of $\theta_{SHE}\lambda_S$ reported for spin Hall materials is 2.5 nm in BiSb[37].

The large $\lambda_{IEE}$ values obtained in our study suggest that the metallicity and the correlated transport with a large Coulomb repulsion of LTO likely play significant roles in spin current transport and in the enhancement of the spin-to-charge conversion efficiency. Note that, in principle, a spin current strongly decays in insulators such as LAO and AlO$_x$, which have been used in previous experiments, even if they are thin[14,15]. Another important factor is the large magnitude of $\alpha_R$. In fact, using $\alpha_R = \lambda_{IEE}\hbar/\tau_e$, we can obtain a large $\alpha_R$ of $8.4 \times 10^{-12}$ eVm for LTO/STO in our experiment (c.f. the reported value of $\alpha_R$ of $3.4 \times 10^{-12}$ eVm for LAO/STO[24]), where $\tau_e$ is obtained from $R_{sheet}$ (see Supplementary Note 3). Furthermore, the well-controlled single-crystalline epitaxial LTO/STO interface, which is atomically flat and sharp owing to MBE growth, seems essential for obtaining a large $\lambda_{IEE}$. Here, we can expect that scattering of the spin current is largely suppressed and that the spin current is efficiently injected into the 2DEG region[4]. In addition, considering that antiferromagnets generally have excellent spin-current propagation[38], the inherent property of the antiferromagnetism of the Mott insulator LTO (Néel temperature is ~150 K)[19] may be somewhat related to the efficient spin injection into the 2DEG through LTO possibly due to the existence of small amount of the antiferromagnetic region.

**Resonant angle-resolved photoemission spectroscopy measurements and theoretical calculations**

To clarify the role of the multiorbital band structure in the IEE, we carry out a theoretical calculation based on an effective tight-binding model[13,39]. We optimize the band parameters so that the calculation reproduces the band dispersion measured by R-ARPES for the reference sample LTO (3 u.c.)/STO shown in Fig. 4a, b. The obtained anisotropic shape of the Fermi surface (Fig. 4b), which is significantly different from a Fermi surface with a $3d^1$ configuration, such as SrVO$_3$[40] and LTO, reflects the following features of the STO interface[29]. While the Ti $d$-orbitals ($d_{xy}, d_{yz}, d_{zx}$) are degenerate in bulk STO, they split due to the 2D confinement at the LTO/STO interface. The $d_{yz}$ and $d_{zx}$ bands have a small effective mass in the $z$ direction due to the spread of the

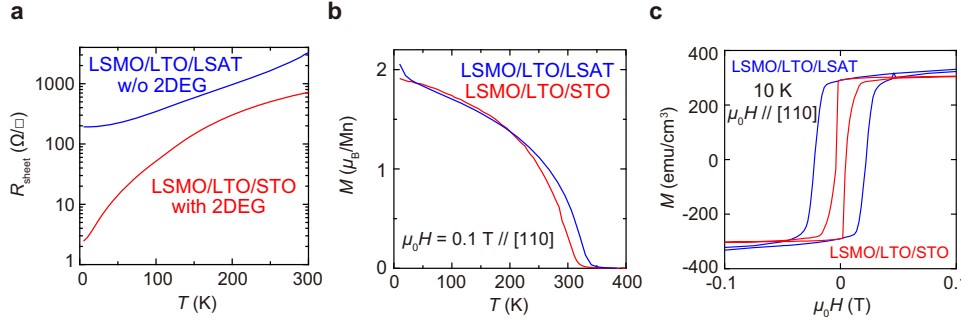

**Fig. 2 | Electrical and magnetization properties of LSMO/LTO/STO and LSMO/LTO/LSAT. a** Temperature ($T$) dependence of the sheet resistance $R_{sheet}$ of the LSMO (30 u.c.)/LTO (3 u.c.)/STO and the LSMO (30 u.c.)/LTO (3 u.c.)/LSAT samples. **b** $T$ dependence of the magnetization $M$ of LSMO (30 u.c.)/LTO (3 u.c.)/STO and LSMO (30 u.c.)/LTO (3 u.c.)/LSAT. The Curie temperatures of LSMO/LTO/STO and LSMO/LTO/LSAT are 320 and 340 K, respectively. **c** Magnetization $M$ for LSMO (30 u.c.)/LTO (3 u.c.)/STO and LSMO (30 u.c.)/LTO (3 u.c.)/LSAT measured at 10 K as a function of a magnetic field $H$ applied along the [110] direction of the STO substrate.

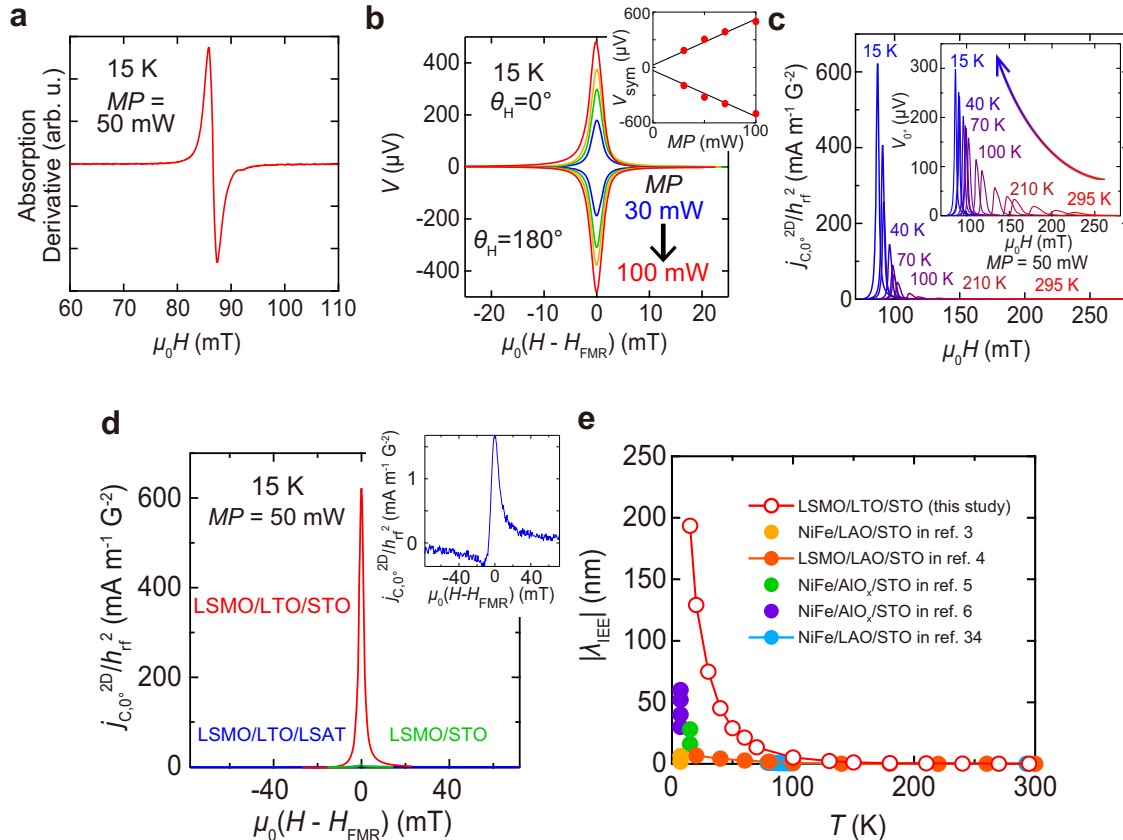

**Fig. 3 | Spin pumping measurements. a** Magnetic field $\mu_0 H$ (// [110] of the STO substrate) dependence of the microwave absorption derivative for the LSMO/LTO/STO sample at 15 K with $MP = 50$ mW. **b** Magnetic field $\mu_0 H$ dependencies of $V$ at 15 K measured for the LSMO/LTO/STO sample with various $MP$ values ranging from 30 to 100 mW. In the electron-spin resonance system, a microwave magnetic field $h_{rf}$ is applied along the [1$\bar{1}$0] direction of the STO substrate. The inset shows the linear relation between the microwave power $MP$ and $V_{sym}$. **c** $H$ dependence of $j_{C,0°}^{2D}/h_{rf}^2$ measured for the LSMO/LTO/STO sample at various temperatures ranging from 15 to 300 K with $MP = 50$ mW. The inset shows the $\mu_0 H$ dependence of $V_{0°}$. **d** Comparison of $j_{C,0°}^{2D}/h_{rf}^2$ between LSMO/LTO/STO, LSMO/LTO/LSAT and LSMO/STO measured with $MP = 50$ mW. The measurements are conducted at 15 K. The inset shows an enlarged figure of the $\mu_0 H$ dependence of $j_{C,0°}^{2D}/h_{rf}^2$ measured for the LSMO/LTO/LSAT sample. **e** Summary of the temperature dependences of $\lambda_{IEE}(= j_{C,sym}^{2D}/j_S^0)$ in various material systems.

wave functions in this direction, leading to large quantization energy so that only the lowest subband is observed below $E_F$ and second and higher quantum levels exist above the Fermi level. Meanwhile, for the $d_{xy}$ band, the wave function is strongly confined in the film plane, and the effective mass is large (small) in the $z$ direction (in-plane $x$ and $y$ directions); thus, the quantization energy is smaller, and the lowest and second subbands appear, as shown in Fig. 4a[29]. Here, the $x$, $y$ and $z$ directions are defined as in-plane [100], in-plane [010] and the perpendicular direction to the film plane (//[001]), respectively. In addition to spin-orbit coupling, the complex hybridization of these bands generates an unconventional Rashba effect with spin splitting (inset in Fig. 4a). The theoretical band dispersion and Fermi surface agree with the R-ARPES results when the carrier density is set to ~3 × 10¹⁴ cm⁻² (Fig. 4a, b) (see Supplementary Note 4). From this calculation result, we

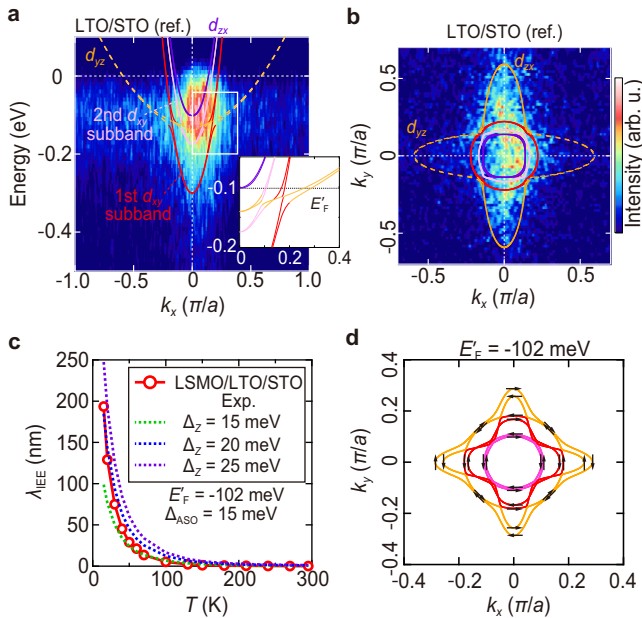

**Fig. 4 | R-ARPES measurement results for the band dispersion and Fermi surface with the theoretical curves and spin orientations, and a comparison between the experimental $\lambda_{IEE}$ and theoretical $\lambda_{IEE}$. a** Band dispersion of a reference sample LTO (3 u.c.)/STO along $k_x$ measured by R-ARPES. The 2DEG states are observed using the resonant photon energy for the Ti $L_3$ absorption edge of the Ti$^{3+}$ component ($h\nu = 459.7$ eV). The solid and broken curves represent the band dispersions calculated by the effective tight-binding model. The inset is the enlarged band dispersion of the calculation near the crossing point of the $d_{xy}$ and $d_{yz}$ bands. The dotted line of the inset corresponds to the Fermi level used for the calculation of $\lambda_{IEE}$ in the LSMO/LTO/STO sample ($E'_F = -102$ meV). Here, $E_F$ and $E'_F$ are the Fermi levels of LTO/STO obtained by the R-ARPES measurement and the Hall effect measurement, respectively (see Supplementary Note 4). **b** Fermi surface of the reference sample LTO (3 u.c.)/STO obtained by R-ARPES. Although it is not clear, the Fermi surface of the thin LTO layer, which is considered to be small due to its small carrier concentration, may be partially overlapped with the Fermi surface of the 2DEG at around the center of the $k_x$–$k_y$ plane. In this measurement, the Fermi surface elongated in the $k_y$ direction is visible, but the surface elongated in the $k_x$ direction is almost not, due to the experimental geometry of the ARPES measurements[48]. The solid and broken curves are the Fermi surfaces calculated by the effective tight-binding model at $E_F = 0$ meV. **c** Temperature $T$ dependence of experimental $\lambda_{IEE}$ for the LSMO/LTO/STO sample and calculated $\lambda_{IEE}$ for $\Delta_z = 15$, 20 and 25 meV with $\Delta_{ASO}$ fixed at 15 meV. **d** Calculated Fermi surface and spin expectation values (arrows) at $E'_F = -102$ meV by the effective tight-binding model.

theoretically derive $\lambda_{IEE}$ (Fig. 4c)[4], where we adjust the magnitude of the spin-orbit coupling $\Delta_{ASO}$ to 15 meV and vary the polar lattice distortion $\Delta_z$. Both parameters influence the effective spin-orbit interaction of this system in this model (see Methods). We set $E_F$ at $E'_F = -102$ meV (see the inset of Fig. 4a and Supplementary Note 4) for all $T$. In Fig. 4c, the calculated results reproduce the experimental $\lambda_{IEE}$ values well. At $E'_F$ ($=-102$ meV), the Fermi surface shows large Rashba splitting (Fig. 4d), which is thought to be one of the main causes of a large $\lambda_{IEE}$. From Fig. 4d, we estimate $\alpha_R = \hbar^2 \Delta k/(2m)$ as $1.0 \times 10^{-11}$ eVm, where we define $\Delta k$ as band spin splitting in the $k_x$ direction at $k_y = 0$ for the first $d_{xy}$ subband. This value of $\alpha_R$ almost agrees with the experimental value of $\alpha_R$ ($8.4 \times 10^{-12}$ eVm) obtained by our spin pumping experiments and the reported values of $\alpha_R$ ($-1.8 \times 10^{-11}$ eVm) obtained by weak antilocalization and theoretical calculation for LTO/STO[18].

## Discussion

For obtaining a large IEE efficiency, we need large $\alpha_R$, large $\tau_e$ and efficient spin-current propagation. When the nonmagnetic (NM) interlayer between the ferromagnetic (FM) layer and STO is insulating, such as in FM/LAO/STO[3,4,34] and FM/AlO$_x$/STO[5,6], most of the converted

charge current flows at the interface between the NM layer and STO. In this case, we can use the large $\tau_e$ of the 2DEG at this interface. However, the spin current injected from the ferromagnetic layer is attenuated when passing through the insulating NM layer before reaching the NM/STO interface[14–16], leading to a decrease in the IEE efficiency. The large loss for the spin current in diamagnetic insulators has been experimentally demonstrated by spin pumping measurements[41]. Meanwhile, when the NM interlayer is purely metallic, the spin current is more efficiently injected into the 2DEG region; however, part of the converted current diffuses to the NM layer, effectively decreasing $\tau_e$ and thus the IEE efficiency[42]. Therefore, choosing an appropriate interlayer material that can maximize both $\tau_e$ and the spin current propagation is crucial for obtaining efficient IEE.

Our result indicates that LTO is a suitable material from this point of view. The spin current can propagate in LTO very efficiently because of its metallicity. Nevertheless, the resistivity $\rho$ of LTO is not so low; $\rho$ is only about $1 \times 10^{-3}$ Ω cm[20], which is much higher than that of ordinary metals. This "moderate" feature of LTO can prevent the reduction in $\tau_e$ of the 2D electrons. Actually, the LSMO/LTO bilayer has a relatively high $R_{sheet}$ of ~200 Ω/□ (on the LSAT substrate) (Fig. 2a). Meanwhile, $R_{sheet}$ of the 2DEG at the LTO/STO interface is ~2 Ω/□, which is two orders of magnitude smaller than that of the LSMO/LTO bilayer as shown in Fig. 2a. Therefore, in our LSMO/LTO/STO sample, most of the current flows at the LTO/STO interface so that $\tau_e$ is not reduced significantly. In addition, the sharpness of the interface in our heterostructures is also thought to substantially increase $\tau_e$ due to the suppression of interface roughness scattering.

As mentioned above, we obtain a large efficiency of spin-to-charge conversion $\lambda_{IEE}$ of up to ~190 nm using 2DEG at the strongly correlated polar-metal LTO/nonpolar STO interface in a high-quality all-epitaxial single-crystal heterostructure, in which we expect that spin scattering is largely suppressed. In contrast to spin Hall systems, our experiments suggest that high-quality samples can lead to high spin-to-charge conversion efficiencies in the case of the IEE. In addition, the Rashba effect at the interface/surface can be designed by using appropriate material combinations for each application. The coexistence of highly efficient spin-current transport and spin-current conversion in LTO/STO makes this material system very attractive. This highly efficient spin transport/conversion highlights the physical phenomena in spin-orbitronics, such as the spin-galvanic effect and spin-orbit torque magnetization switching, and paves the way for ultralow-power computing and storage by spintronic devices.

## Methods

### Sample preparation

The samples used in this study were grown on TiO$_2$-terminated STO (001) substrates and LSAT (001) substrates by ultrahigh-vacuum MBE. Prior to growth, to obtain the TiO$_2$-terminated surface, the STO substrates were etched with buffered hydrofluoric acid (HF) for 30 s and annealed at 1000 °C for 1 h under ambient conditions. The LSAT substrates were also annealed at 1000 °C for 1 h under ambient conditions. We calibrated the La, Ti, Sr and Mn fluxes by integrating each flux value for 30 min using a quartz-crystal microbalance-thickness monitor placed at the substrate position in the MBE chamber. We used a shuttered growth technique with fluxes supplied from pure metallic La, Ti, Sr and Mn sources in Knudsen cells. The LTO layer was grown at 600 °C with a background pressure of $2 \times 10^{-7}$ Pa without introducing oxygen O$_2$ or ozone O$_3$[20,23]. The first 2 u.c. of the LSMO layer were grown at 600 °C with a background pressure of $2 \times 10^{-7}$ Pa. The next 13 u.c. of the LSMO layer were grown at 730 °C with a background pressure of $5 \times 10^{-5}$ Pa due to a mixture of O$_2$ (80%) and O$_3$ (20%). The remaining 15 u.c. of the LSMO layer were grown at 730 °C with a background pressure of $2 \times 10^{-4}$ Pa due to the same mixture of O$_2$ and O$_3$. These processes can prevent the structural change from perovskite-phase LTO to pyrochlore-phase La$_2$Ti$_2$O$_7$ (Supplementary

Fig. 3)[43–45]. The RHEED patterns during the growth of LTO and LSMO are shown in Supplementary Fig. 12.

### Preparation of the Hall bar and electrical transport measurements

We made a Hall bar with a size of $100 \times 400\ \mu m^2$ by standard photolithography and Ar ion milling. Then, contact pads were formed on the Hall bar using sputter deposition and a lift-off process of a 100 nm-thick Al film. The transport measurements were conducted by a standard four-terminal method using a Quantum Design physical property measurement system.

### Spin pumping measurements

We carried out spin pumping measurements using a transverse electric (TE$_{011}$) cavity of an electron-spin resonance system with a microwave frequency of 9.1 GHz in an electron-spin resonance system JES-FA300, JEOL. We cut the samples into a small piece with a size of $1.4 \times 0.95$ mm, connected gold wires to the contacts at both edges of the sample (electrode distance $l = 1.2$ mm), and placed the sample at the center of the cavity. To take the measurements, a static magnetic field $\mu_0 H$ was applied along the [110] direction in the film plane, which corresponds to the easy magnetization axis of LSMO (see ref. 4 and Supplementary Fig. 6) and can suppress the contribution of the planer Hall effect[33]. Meanwhile, a microwave magnetic field $h_{rf}$ was applied along the [1$\bar{1}$0] direction. The microwave power used was fixed at 50 mW in the temperature-dependence experiments.

We estimated the spin current density $j_S^0$ from the mixing conductance $g_r^{\uparrow\downarrow}$. Here, $g_r^{\uparrow\downarrow}$ was obtained from the difference in the damping constant $\alpha = \frac{\sqrt{3}\gamma \triangle H_{pp}}{2\omega}$ ($\triangle H_{pp}$: FMR peak-to-peak linewidth) between LSMO/LTO/STO (Fig. 3a) and LSMO/STO[32] as follows[46]:

$$g_r^{\uparrow\downarrow} = \frac{4\pi M_S d_{LSMO}}{g\mu_B}\left(\frac{\sqrt{3}\gamma}{2\omega}\triangle H_{pp,LSMO/LTO/STO} - \frac{\sqrt{3}\gamma}{2\omega}\triangle H_{pp,LSMO}\right) \quad (1)$$

where M$_S$, $d_{LSMO}$, $g$, $\mu_B$, $\triangle H_{pp,LSMO/LTO/STO}$, $\triangle H_{pp,LSMO}$, $\gamma$ and $\omega$ are the saturation magnetization of the LSMO, thickness of the LSMO layer, $g$-factor[47], Bohr magneton, FMR peak-to-peak linewidth of the LSMO/LTO/STO sample, FMR peak-to-peak linewidth of the LSMO/STO sample, gyromagnetic ratio and angular frequency, respectively. Here, we set the damping constant of $\frac{\sqrt{3}\gamma}{2\omega}\triangle H_{pp,LSMO}$ as $1.57 \times 10^{-3}$, which was reported for a high-quality LSMO film in ref. 32. Since $\frac{\sqrt{3}\gamma}{2\omega}\triangle H_{pp,LSMO}$ is thought to depend on the crystal quality, this choice means that we may overestimate $j_S^0$ and thus underestimate $\lambda_{IEE}$. $j_S^0$ injected into LTO is expressed as:

$$j_S^0 = \frac{g_r^{\uparrow\downarrow}\gamma^2 h_{rf}^2 \hbar\left[4\pi M_S\gamma + \sqrt{(4\pi M_S)^2\gamma^2 + 4\omega^2}\right]}{8\pi\alpha^2\left[(4\pi M_S)^2\gamma^2 + 4\omega^2\right]} \quad (2)$$

where $h_{rf}$ is the microwave magnetic field. The obtained values of $\alpha$, $g_r^{\uparrow\downarrow}$ and $j_S^0$ are shown in Supplementary Fig. 7. Before measuring each sample, we checked the quality factor $Q$ with the sample inserted into the cavity and estimated the $h_{rf}$ value using the data of $h_{rf}$ vs. $Q$ provided by JEOL.

### Resonant angle-resolved photoemission spectroscopy

R-ARPES measurements were performed in an ultrahigh vacuum below $1 \times 10^{-8}$ Pa at 30 K with a Scienta Omicron SES-2002 electron energy analyzer at beamline BL-2A MUSASHI of the Photon Factory, KEK. The total-energy resolution was set to 150 meV at the incident photon energy $h\nu$ of around 400 eV. The R-ARPES spectra were recorded using linear horizontal light and horizontal detection slit geometry, resulting in observation of electronic states derived mainly from the $d_{zx}$ orbital (and slightly from the $d_{xy}$ and $d_{yz}$ orbitals). To avoid surface contamination, we transferred the sample to a measurement chamber in a

vacuum suitcase filled with N$_2$ gas without exposure to air. By selecting the incident photon energy at the Ti$^{3+}$ $L_3$ edge ($h\nu = 459.7$ eV), the signal of 2D conductivity at the LTO/STO interface was greatly enhanced[48,49]. The Fermi level of the samples was referred to that of a gold foil that was in electrical contact with the sample. The Fermi surface map (Fig. 4b) was obtained by plotting the ARPES intensity within the energy window of ±25 meV from the Fermi level.

### Calculation of $\lambda_{IEE}$

To theoretically estimate $\lambda_{IEE}$, we calculated the band structure of the 2DEG region at the STO interface using the effective tight-binding model. The Hamiltonian used in the calculation is expressed below, where we add the second Ti $d_{xy}$ subband to the Hamiltonian proposed in ref. 39. In what follows, $H_0$ represents electrons hopping between neighboring atoms and on-site interactions, $H_{ASO}$ is the term for the spin-orbit interaction of atoms, and $H_a$ is the term for the polar lattice distortion caused by the electric field due to the broken inversion symmetry (linear Rashba-like term):

$$H_0 = \begin{pmatrix} \frac{\hbar^2 k_x^2}{2m_h} + \frac{\hbar^2 k_y^2}{2m_l} & 0 & 0 & 0 \\ 0 & \frac{\hbar^2 k_x^2}{2m_l} + \frac{\hbar^2 k_y^2}{2m_h} & 0 & 0 \\ 0 & 0 & \frac{\hbar^2 k_x^2}{2m_l} + \frac{\hbar^2 k_y^2}{2m_l} - \triangle_{E1} & 0 \\ 0 & 0 & 0 & \frac{\hbar^2 k_x^2}{2m_l} + \frac{\hbar^2 k_y^2}{2m_l} - \triangle_{E2} \end{pmatrix} \otimes \sigma^0, \quad (3)$$

$$H_{ASO} = \triangle_{ASO}\begin{pmatrix} 0 & i\sigma_z & -i\sigma_y & -i\sigma_y \\ -i\sigma_z & 0 & i\sigma_x & i\sigma_x \\ i\sigma_y & -i\sigma_x & 0 & 0 \\ i\sigma_y & -i\sigma_x & 0 & 0 \end{pmatrix}, \quad (4)$$

$$H_a = \triangle_z\begin{pmatrix} 0 & 0 & ik_x & ik_x \\ 0 & 0 & ik_y & ik_y \\ -ik_x & -ik_y & 0 & 0 \\ -ik_x & -ik_y & 0 & 0 \end{pmatrix} \otimes \sigma^0, \quad (5)$$

where $\hbar$ is the Dirac constant, $k_x$ and $k_y$ are the wavenumbers in the $x$ and $y$ directions, respectively, $\Delta_{E1}$ ($\Delta_{E2}$) is the band splitting between the first (second) $d_{xy}$ subband and $d_{yz}$ band due to the confinement of the wave function in the $z$ direction, and $\Delta_{ASO}$ and $\Delta_z$ express the coefficients of $H_{ASO}$ and $H_a$, respectively. $\sigma^0$ is the identity matrix in spin space, and $\sigma_x$, $\sigma_y$ and $\sigma_z$ are the spin matrices. $\otimes$ is the Kronecker product. $m_h = 6.8\ m_0$ and $m_l = 0.41\ m_0$ are the effective masses of the heavy and light electrons in STO[50], and $m_0$ is the free electron mass. The Hamiltonian $H = H_0 + H_{ASO} + H_a$ is an $8 \times 8$ matrix due to the up- and down-spins. We adjusted the energy position (i.e., the energy distance from the band bottom to $E_F$) of the entire band structure so that the Fermi surface width of the $d_{zx}$ band in the $k_y$ direction matches the one obtained by the R-ARPES (see Fig. 4b).

Next, we calculated $\lambda_{IEE}$ using the above Hamiltonian[4,13]. $\lambda_{IEE}$ is expressed as $j_C^{2D}/j_S^0$, where the two-dimensional current $j_C^{2D}$ and $j_S^0$ are derived from the following equations:

$$j_C^{FSn} = \frac{e^2}{4\pi^2\hbar}\int^{FS_n} F_x(\mathbf{k})dS_F \quad (6)$$

$$F_x(\mathbf{k}) = F\,\mathrm{sgn}\left(S_y(\mathbf{k})\right)\tau(|\mathbf{k}|)v_x(\mathbf{k})\frac{v_x(\mathbf{k})}{|\mathbf{v}(\mathbf{k})|} \quad (7)$$

$$\delta s^{FSn} = \frac{e}{4\pi^2\hbar}\int^{FS_n}\int |S_y(\mathbf{k})|dS_F \quad (8)$$

$$S_y(\mathbf{k}) = F\tau(|\mathbf{k}|)\sigma_y(\mathbf{k})\frac{v_x(\mathbf{k})}{|\mathbf{v}(\mathbf{k})|} \quad (9)$$

$$j_C^{2D} = \sum_n j_c^{FSn}, j_S^0 = \sum_n \frac{e\delta s^{FSn}}{\tau_e} \qquad (10)$$

where $F$ is the electric field, $\delta s^{FSn}$ is the spin accumulation, $\tau(|\mathbf{k}|)$ is the relaxation time of electrons at each $\mathbf{k}$ point, and $e$ is the elementary charge[51]. $\mathbf{v}(\mathbf{k}) = (v_x(\mathbf{k}), v_y(\mathbf{k}))$ and $S_y$ are the group velocity and the spin magnitude in the $y$ direction, respectively. $n$ is the index of each band. The integrations in Eqs. (6) and (8) are conducted over each Fermi surface $FS_n$. $\tau(|\mathbf{k}|)$ is assumed to be proportional to $|\mathbf{k}|$.

## Data availability

Source data are provided with this paper.

## Code availability

The code that generated the data for Fig. 4 is available from the corresponding author on reasonable request.

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

## Acknowledgements

This work was partly supported by Grants-in-Aid for Scientific Research (Nos. 18H03860, 18K14130, 20H05650, 21J21102, 21K14541, 22H04948), CREST (No. JPMJCR1777) and PRESTO (No. JPMJPR19LB) of the Japan Science and Technology Agency, the Spintronics Research Network of Japan (Spin-RNJ), and the ANRI fellowship. Part of this work was conducted at the Advanced Characterization Nanotechnology Platform of the University of Tokyo, and it was supported by the "Nanotechnology Platform" of the Ministry of Education, Culture, Sports, Science and Technology (MEXT), Japan, and the Cryogenic Research Center of the University of Tokyo. S.K.T. acknowledges support from the Japan Society for the Promotion of Science (JSPS) Fellowships for Young Scientists. The work performed at Photon Factory, KEK was approved by the Program Advisory Committee (proposals 2019G600 and 2021G644) at the Institute of Materials Structure Science, KEK.

## Author contributions

Experimental design, growth and data analysis: S.K.T.; device fabrication and measurements: S.K.T., T.A. and T.E.; R-ARPES measurements: S.K.T., Mi.K., R.O., K.H., H.K. and Ma.K.; theoretical calculation: S.K.T., S.A. and S.O.; writing and project planning: S.K.T., M.T. and S.O.; S.K.T., Mi.K., L.D.A., Ma.K., M.S., H.T., M.T. and S.O. discussed the results and the manuscript extensively.

## Competing interests

The authors declare no competing interests.
