## [Peer Review File · Nature Communications]

REVIEWER COMMENTS

Reviewer #1 (Remarks to the Author):

The authors studied the spin to charge current conversion in an all epitaxial single crystal oxide Rashba interface. They used spin pumping by ferromagnetic resonance in a LSMO/LaTiO₃+ δ /STO heterostructure, injecting spin current from the ferromagnetic LSMO towards the two-dimensional electron gas at the LaTiO₃+ δ /STO (LTO/STO) interface.

The two-dimensional electron gas (2DEG) at the interface/surface of STO is already well known for its very high and gate tunable spin to charge current conversion as previously demonstrated in FM/LaAlO₃/STO and FM/AlO_x/STO for example (reference 2,3,4,5,24 and 28 of the manuscript). Some authors of this paper already published an interesting article on the conversion in all epitaxial LSMO/ LaAlO₃/STO in Physical Review Research (reference 3). The main novelty of this study is the use of conducting strongly correlated polar metal LTO between the ferromagnetic spin current injector and the 2DEG, while previous studies used insulating LaAlO₃ and AlO_x. Using this LTO layer they obtained a conversion efficiency considerably larger than LSMO/ LaAlO₃/STO and associate it with the larger Rashba splitting at the LTO/STO interface, the metallic nature of the interface and the strong on-site Coulomb repulsion in conductive LTO.

These results are of significance in the field of oxide spinorbitronics in particular the use of LTO appears as an alternative for spintronics applications because of its peculiar properties and of the larger Rashba splitting at the LTO/STO interface. STO based 2DEGs are systems with multiple properties (magnetism, superconductivity...etc) and the use of a possibly conductive overlayer in contact with the 2DEG is to the best of my knowledge new, that's why these results are not only interesting for the field of oxide spinorbitronics but are of enough broad interest to be published in Nature Communications.

Nonetheless I think this manuscript has to be revised in particular concerning the electrical properties of the LTO and of the 2DEG. My main concern being on the nature of the the transport at the interface with STO, the exact carrier density, the metallicity of the LTO layer and the role of the metallicity or antiferromagnetism in the large spin pumping signal. I have also some more minor comments on other parts of the manuscript.

Carrier density:

From figure 2a) it is clear that the sheet resistance at low temperature is particularly low, of the order of $2 \Omega/\text{square}$. This value is considerably lower than previously reported in LTO/STO (see Biscaras, J. et al. Sci Rep 4, 6788 (2014) for example) and in other STO based 2DEGs systems with carrier density in the 10^{13} cm^{-2} range. The parallel conduction in the LSMO and LTO (of around $200 \Omega/\text{square}$) cannot explain such a low sheet resistance. This very low sheet resistance appears incompatible with the carrier density used to explain the large conversion efficiency of $4 \times 10^{13} \text{ cm}^{-2}$ (supplementary figure S7). In their previous publication (reference 3 of the manuscript), the authors obtained a similar sheet resistance for a 2DEG with a much higher carrier density of $2 \times 10^{14} \text{ cm}^{-2}$ (in sample ref A).

Can the authors comment on that apparent discrepancy?

Can the author perform a Hall measurement at low temperature to estimate the carrier density in the LSMO/ LTO/STO to confirm the carrier density?

An accurate estimation of the carrier density should be given to better understand the low sheet resistance and to have a more accurate understanding of the position of the Fermi level that is essential for the theoretical evaluation of the inverse Edelstein length?

Nature of the transport in STO:

The authors state that they have a 2DEG but due to the low sheet resistance and high temperature deposition it is unclear if it is the case and the ARPES data are not very clear. The authors deposit LTO at a high temperature of 600°C and a low oxygen pressure of $2 \times 10^{-7} \text{ Pa}$, such high temperature and low-pressure growth is known to induce oxygen vacancies as seen for example in Phys. Rev. Lett. 98, 216803 (2007) or EPL 91, 17004 (2010) for depositions at higher temperatures of 750°C and 800°C . These oxygen vacancies extend deeply in the bulk of STO (see Nature Mater 7, 621–625 (2008)) and lead to a low sheet resistance, high mobility and higher carrier density.

Are the STO substrates still insulating after a similar treatment without deposition of the LTO?

Do the authors have any experimental evidence of the two-dimensional nature of the transport in LTO/STO deposited using their technique?

Metallicity of the LTO:

The authors state that the LTO they use is conductive due to the excess of oxygen. Nonetheless they show no evidence of the metallicity of the LTO in this work. I understand it could be difficult in the

LSMO/LTO/STO sample as LTO is only 3 uc but it could be simpler in a bilayer of LTO/STO with thick LTO where the contribution of LTO is dominant as shown in Phys. Rev. B 81, 161101 (2010).

In the supplementary information the authors present a sample with thick LTO (20 uc) on STO (figure S3), can they confirm that in this sample the LTO is metallic and its contribution to the transport is dominant?

In reference 7 and Phys. Rev. B 81, 161101 (2010) the authors associate the conductivity of LTO to the strain due to epitaxial deposition on STO while the authors associate it with oxygen excess. Can the authors explain why they consider the oxygen excess and not the strain?

Role of the metallicity of the LTO:

The authors observe a very high conversion efficiency and associate it with the metallicity of the LTO layer. They write: "The large λ_{IEE} values obtained in our study suggest that the metallicity and the correlated transport with a large Coulomb repulsion of LTO likely play significant roles in spin current transport and in the enhancement of the spin-current conversion efficiency." But in the case of the inverse Rashba Edelstein effect, the spin to charge conversion should be higher with an insulating interface as shown for example by Fert and Zhang in Phys. Rev. B 94, 184423 (2019). Indeed, as stated by the authors the conversion efficiency is proportional to the momentum relaxation time of the electrons τ_e , with $\lambda_{IEE} = \alpha R^* \tau_e / \hbar$, and a conductive interface would reduce the momentum relaxation time as an additional relaxation channel.

Can the authors comment on that apparent discrepancy between previous works, for spin to charge conversion an insulating interface is favorable, and their statement on conductive interfaces?

Role of the antiferromagnetism in LTO:

The authors state that "antiferromagnets generally have excellent spin-current propagation, the inherent property of the antiferromagnetism of the Mott insulator LTO may be somewhat related to efficient spin injection into the 2DEG through LTO." It is indeed true that insulating antiferromagnets, have excellent spin-current propagation properties, for example NiO. especially close to the Néel temperature as seen in Phys. Rev. Lett. 116, 186601

(2016).

Can the author comment on the Néel temperature of LTO? Is it compatible with highly efficient spin current injection?

Comments on the ARPES data:

I am not a specialist of ARPES but it seems that the quality of the ARPES data is not good enough to clearly observe the bandstructure of LTO/STO and the calculated bandstructure doesn't look like the experimental data. It is especially unclear to me how it is possible to observe the bandstructure of 2DEG in STO without any additional contribution of the LTO if it is conductive (states at the Fermi level in LTO should appear).

Can the authors comment on the differences between the ARPES and the calculated bandstructure?

Can the authors comment on the absence of the d-states of conductive LTO in the ARPES?

To sum up in its current form the manuscript is unclear on the exact electrical properties of LTO (insulating or conductive) and of the 2DEG properties. The role of the metallicity or antiferromagnetism of LTO also have to be clarified to better understand why such a high conversion efficiency is obtained.

Other comments:

-The authors want to have LTO with oxygen in excess but deposit LTO at very low oxygen pressure, how is it compatible?

-In the methods section it is unclear if the targets used are metallic or oxides, can the author clarify this point. If metallic can the author comment on the possible role of Ti in the formation of oxygen vacancies?

-The authors do not show any data on the damping versus the temperature contrary to their previous work. It would be interesting to have these data as well as the damping versus temperature in their reference sample for comparison purposes. The estimated spin current injected and spin mixing conductance should also be shown.

-The authors use LSMO/LTO/LSAT as their reference sample. As LSAT has very different dielectric properties compared with STO I am not sure that is the ideal reference sample. Due to its large dielectric constant and dielectric loss STO can slightly affect the cavity properties and lead to enhanced rectification effects. Do the authors have similar spin pumping FMR measurements in LSMO/STO?

-The authors write "To extract a pure IEE signal, we decompose the EMF into a symmetric (Lorentzian) component V_{sym} , which includes signals of the IEE, and an antisymmetric (anti-Lorentzian) component V_{asym} ." But this is generally not true the rectification effects can have a symmetric and an antisymmetric component as shown in Phys. Rev. B 88, 064403 (2013) for example. Ideally an angular dependence measurement should be performed. As the author

observed no effect in a similar sample without a 2DEG the contribution of the rectification effects should be small.

-In figure 4c) when the spin orbit interaction decreases the conversion efficiency increases, this is very counter intuitive can the authors comment on this?

-In the introduction the authors wrote “Moreover, the mechanism of the spin-current flow in these insulating layers is still unexplored.” I would like to point out that a recent article studies this mechanism Phys. Rev. Research 3, 043170 (2021).

Reviewer #2 (Remarks to the Author):

In this work by Kaneta-Takada et al, the authors show a large spin pumping voltage signal in a 2DEG system based on the over oxygenated polar metal LaTiO₃ on SrTiO₃ substrate. As an injector layer, they use 12 nm of the LSMO manganite, which is also epitaxial. The voltage reaches values as high as 500 μ V at 15 K! As the sample is poorly resistive at 15 K, due to the high conductivity of the 2D gas and the metallic character of the doped LaTiO₃, the estimated efficiency turns out to be giant. The normalized value of the two-dimensional charge current production exceeds 600 mA/mG², and the efficiency reaches values of 190 nm. A record. In addition, the authors show ARPES measurements with theoretical fit highlighting the nature of the two-dimensional gas. This result shows a new system very attractive to the community and opens new possibilities for further studies of spin-charge interconversion as well as potential applications. One would like to know for example about the anisotropy, or not, of the interconversion, as well as a dependence on the gate voltage. Surely that will be done in future work. The measurements are convincing, indeed the authors show a very attractive new system, with record values of spin current to charge current conversion. Therefore, my advice is to recommend the publication of this manuscript.

The reviewer has some minor details to point out:

In the abstract, please add that the demonstration of giant spin-to-charge current conversions efficiencies is by Spin pumping FMR voltage measurements.

Line 73-75: if there is a spin texture, such as spin-momentum locking, and which is schematized in Figure 4d, the relaxation time indicated in the equation on line 73 is not of the electron momentum, or of the spin relaxation time, but of both.

How did the authors measure or estimate h_{rf} ? h_{rf} is used in equation (2). I have read reference (3) from the same group but have not found information on how the authors estimate h_{rf} . This parameter, which depends on the experimental setup, is important. A small error in h_{rf} will strongly impact the estimation of J_s and consequently the interconversion efficiency. Can the authors also indicate the model of their spectrometer? If the authors use h_{rf} given by the fabricator, that would be for an empty cavity. But when the sample is inserted, it changes its quality factor Q and thus the h_{rf} . Do the authors consider that h_{rf} does not change when the sample is inserted into the resonant cavity?

I suppose that the measured voltage curves, electromotive force EMF in the authors' notation, are the raw data, right? It seems so from looking at the curves in Figure 3b and 3d. However, if so the authors should state it explicitly, or the authors show only the symmetrical contribution?

In line 26, it appears that the authors made a mistake or it is a typo. The 2D charge current is the measured spin pumping voltage normalized by the total sample resistance and the bar width (0.95 mm), not the distance between the electrodes (1.2 mm) as stated in the manuscript. Please revise that.

In the present work, the authors perform their measurements when H is applied along the easy axis of in-plane magnetic anisotropy, $[110]$, and the voltage is measured at 90 degrees from that direction. Have the authors measured when H is applied in another direction? Could they comment on this? Have they chosen the easy axis because the resonance field would be lower, or because the efficiency is higher in this direction?

Response letter

We would like to thank the reviewers for valuable comments, which helped us improve the quality of our manuscript (NCOMMS-22-00119-T). We have carefully revised our manuscript following those comments. In this Response letter, we reply to all the comments, point by point, and describe how we revised the manuscript. Here, the comments of the reviewers are written in blue color. The revised parts are written in red color in this Response letter, the main manuscript and the Supplemental Information.

Reviewer #1's comments

The authors studied the spin to charge current conversion in an all epitaxial single crystal oxide Rashba interface. They used spin pumping by ferromagnetic resonance in a LSMO/LaTiO_{3+δ}/STO heterostructure, injecting spin current from the ferromagnetic LSMO towards the two-dimensional electron gas at the LaTiO_{3+δ}/STO (LTO/STO) interface.

The two-dimensional electron gas (2DEG) at the interface/surface of STO is already well known for its very high and gate tunable spin to charge current conversion as previously demonstrated in FM/LaAlO₃/STO and FM/AIO_x/STO for example (reference 2,3,4,5,24 and 28 of the manuscript). Some authors of this paper already published an interesting article on the conversion in all epitaxial LSMO/ LaAlO₃/STO in Physical Review Research (reference 3). The main novelty of this study is the use of conducting strongly correlated polar metal LTO between the ferromagnetic spin current injector and the 2DEG, while previous studies used insulating LaAlO₃ and AIO_x. Using this LTO layer they obtained a conversion efficiency considerably larger than LSMO/ LaAlO₃/STO and associate it with the larger Rashba splitting at the LTO/STO interface, the metallic nature of the interface and the strong on-site Coulomb repulsion in conductive LTO.

These results are of significance in the field of oxide spinorbitronics in particular the use of LTO appears as an alternative for spintronics applications because of its peculiar properties and of the larger Rashba splitting at the LTO/STO interface. STO based 2DEGs are systems with multiple properties (magnetism, superconductivity...etc) and the use of a possibly conductive overlayer in contact with the 2DEG is to the best of my knowledge new, that's why these results are not only interesting for the field of oxide spinorbitronics but are of enough broad interest to be published in Nature Communications.

Nonetheless I think this manuscript has to be revised in particular concerning the electrical properties of the LTO and of the 2DEG. My main concern being on the nature of the the transport at the interface with STO, the exact carrier density, the metallicity of the LTO layer and the role of the metallicity or antiferromagnetism in the large spin pumping signal. I have also some more minor comments on other parts of the manuscript.

Comment 1:

Carrier density:

From figure 2a) it is clear that the sheet resistance at low temperature is particularly low, of the order of 2 Ω /square. This value is considerably lower than previously reported in LTO/STO (see Biscaras, J. *et al.* Sci Rep **4**, 6788 (2014) for example) and in other STO based 2DEGs systems with carrier density in the 10^{13} cm^{-2} range. The parallel conduction in the LSMO and LTO (of around 200 Ω /square) cannot explain such a low sheet resistance. This very low sheet resistance appears incompatible with the carrier density used to explain the large conversion efficiency of 4×10^{13} cm^{-2} (supplementary figure S7). In their previous publication (reference 3 of the manuscript), the authors obtained a similar sheet resistance for a 2DEG with a much higher carrier density of 2×10^{14} cm^{-2} (in sample ref A). Can the authors comment on that apparent discrepancy? Can the author perform a Hall measurement at low temperature to estimate the carrier density in the LSMO/LTO/STO to confirm the carrier density?

Our response:

We would like to thank the reviewer for the positive evaluation of our manuscript and the valuable kind comment. Following it, we measured the sheet carrier density n_{sheet} and the sheet resistance R_{sheet} of the $\text{LaTiO}_{3+\delta}$ [3 unit-cell (u.c.)]/ SrTiO_3 (STO) heterostructure, which we found to be $\sim 8.9 \times 10^{13}$ cm^{-2} (by the Hall effect) and ~ 8 Ω/\square at 4 K, respectively. The reason for the difference between n_{sheet} estimated by the Hall effect and by resonant angular-resolved photoemission spectroscopy (R-ARPES, the estimated n_{sheet} is $\sim 3 \times 10^{14}$ cm^{-2}) may be due to the oxidation of the $\text{LaTiO}_{3+\delta}$ /STO sample used for the Hall effect (the sample used for the APRES measurement was transferred from the molecular beam epitaxy (MBE) chamber to the ARPES chamber under a N_2 -gas atmosphere, so that the oxidization of the sample was suppressed).

A wide range of n_{sheet} and R_{sheet} has been reported for the two-dimensional electron gas (2DEG) at STO interfaces. In Fig. R1, we summarize R_{sheet} vs. n_{sheet} reported for the 2DEG at various STO interfaces grown by sputtering, pulse laser deposition (PLD)

and MBE. As indicated by the green points obtained for LaAlO₃ (LAO)/STO, in comparison with the samples grown by PLD (see the open green points), lower R_{sheet} can be realized by using MBE (see the filled green points). This trend is probably due to the suppression of the interface roughness scattering. By using MBE, as demonstrated in our previous publication of ref. 3 in the manuscript [Ohya, S. et al. *Phys. Rev. Res.* **2**, 012014 (R) (2020)], we obtained the 2DEG with R_{sheet} of $2 \Omega/\square$ with n_{sheet} of $\sim 2 \times 10^{14} \text{ cm}^{-2}$ at the LAO/STO interface, and we also achieved high-quality LAO/STO with R_{sheet} of $\sim 8 \Omega/\square$ and n_{sheet} of $\sim 5 \times 10^{13} \text{ cm}^{-2}$ (not published). We note that the 2DEG at the γ -Al₂O₃ (GAO)/STO interface showed a low R_{sheet} of $\sim 1.5 \Omega/\square$ with n_{sheet} of $\sim 8.5 \times 10^{13} \text{ cm}^{-2}$, even though PLD was used for the growth [see the blue open points. Christensen, D. V. et al. *Phys. Rev. Appl.* **9**, 054004 (2018)]. The light orange zone in Fig. R1 expresses the range of R_{sheet} and n_{sheet} reported for various STO interfaces. Our data of R_{sheet} of $\sim 2.5\text{--}8 \Omega/\square$ with $n_{\text{sheet}} \sim 8.9 \times 10^{13} \text{ cm}^{-2}$ obtained in our LSMO/LaTiO_{3+ δ} /STO heterostructure (see the dark orange region) are well within this range.

Fig. R1. R_{sheet} VS. n_{sheet} at various STO interfaces. The red, green, blue and light blue points correspond to the data of LaTiO_{3+ δ} /STO, LAO/STO, GAO/STO and AlO_x/STO, respectively. The filled and open points are the data of samples made by MBE and PLD, respectively (only the light blue point corresponds to a sample made by sputtering). The light orange zone is the range of R_{sheet} and n_{sheet} for various STO interfaces reported so far [1]–[12]. The dark orange zone is the range obtained in our LSMO/LaTiO_{3+ δ} /STO heterostructure.

References:

[1] Biscaras, J. et al. *Sci. Rep.* **4**, 6788 (2014). [2] Lee, J. N. et al. *Appl. Phys. Lett.* **116**, 171601 (2020). [3] Veit, M. J. et al. *Nat. Commun.* **9**, 1458 (2018). [4] Lesne, E. et al. *Nat. Mater.* **15**, 1261 (2016). [5] Chen, Z. et al. *Nano. Lett.* **16**, 6130 (2016). [6] Chen, Y. Z. et al. *Nat. Commun.* **4**, 1371 (2013). [7] Christensen, D. V. et al. *Phys. Rev. Appl.* **9**, 054004 (2018). [8] Vaz, D. C. et al. *Nat. Mater.* **15**, 1187 (2019). [9] Kaneta-Takada, S. in the revised manuscript. [10] Edmondson, B. I. et al. *J. Appl. Phys.* **124**, 185303 (2018). [11] Sun, H. Y. et al. *Nat. Commun.* **9**, 2965 (2018). [12] Ohya, S. et al. *Phys. Rev. Res.* **2**, 012014 (R) (2020) and not published data.

➤ **Corresponding revised parts in the manuscript:**

- ◇ Revisions related to this response are summarized after the response to the following Comment 2.

Comment 2:

An accurate estimation of the carrier density should be given to better understand the low sheet resistance and to have a more accurate understanding of the position of the Fermi level that is essential for the theoretical evaluation of the inverse Edelstein length?

Our response:

As mentioned above, we estimated the carrier density of the $\text{LaTiO}_{3+\delta}$ (3 u.c.)/STO heterostructure to be $\sim 8.9 \times 10^{13} \text{ cm}^{-2}$ at 4 K. We found that the Fermi level estimated from this value (E'_F) is 0.102 eV lower than that obtained by ARPES (see Supplementary Fig. 9 in the revised Supplementary Information. *i.e.* $E'_F = -0.102 \text{ eV}$).

We conducted the theoretical calculation of the inverse Edelstein length λ_{IEE} by using this E'_F value (Fig. R2). Here, we treated the coefficient Δ_z of the polar lattice distortion H_a also as a variable parameter as well as the spin-orbit coupling Δ_{ASO} [both parameters express the spin orbit interaction. For more details, see Kim, Y. et al. *Phys. Rev. B* **87**, 245121 (2013)]. We have found that the calculated results well reproduce the experimental λ_{IEE} value at 15 K (193.5 nm) when Δ_z is 20 meV and Δ_{ASO} is 15 meV.

Fig. R2. E_F dependence of calculated λ_{IIIIE} when Δ_{ASO} is fixed at 15 meV. The crossing point of the dotted lines corresponds to the experimental value of λ_{IIIIE} and the E_F ($= E'_F = -0.102$ eV) estimated by the Hall effect, where the carrier density is $\sim 8.9 \times 10^{13} \text{ cm}^{-2}$.

➤ **Corresponding revised parts in the manuscript:**

- ✧ We revised the calculated band structure, Fermi surface, Fermi energy, and theoretical λ_{IIIIE} in Fig. 4a,b,c,d of the revised manuscript.
- ✧ We added the description about the coefficient Δ_Z in line 204–208 on page 9 in the revised manuscript.
- ✧ We replaced the data of the calculated sheet carrier density, band structure, Fermi surface, and Fermi level in Supplementary Fig. 9 and 10 in the revised Supplementary Information.
- ✧ We changed the description about the Fermi level on page 10–11 in the revised Supplementary Information:

- (Old)
1. To reproduce the resonant angle-resolved photo emission spectroscopy (R-ARPES) results, we set the parameters $\Delta_{E1} = 0.12$ eV, $\Delta_{E2} = 0.02$ eV, $\Delta_z = 0.01$ eV in the effective tight-binding model (see each definition in the Methods). In addition, we set $\Delta_{ASO} = 0.015$ eV so that the experimental λ_{IIIIE} can be reproduced by the calculation.
 2. We determine E_F so that the temperature dependence of the experimental λ_{IIIIE} agrees with the calculated values, where E_F is fixed and independent of temperature. Only the momentum relaxation time τ_c of electrons [see Sec. III in this Supplementary information and Supplementary Fig. 6], which is derived from the experimentally obtained sheet resistance R_{sheet} , depends on the temperature and

causes the temperature dependence of the λ_{IEE} in our model.

For the LSMO/LTO/STO sample, we use the value of the E_{F} determined in the above procedure (*i.e.* $E_{\text{F}} = E'_{\text{F}} = -0.137$ eV in Fig. 4 in the main text) for the calculation of n_{sheet} . For the LTO/STO sample, the Fermi level is measured to be located at ~ 0.254 eV from the 1st d_{xy} subband bottom from the R-ARPES experiment (see Fig. 4a in the main text). Thus, we calculate n_{sheet} for LTO/STO by integrating the density of states from -0.254 eV to 0 eV in Supplementary Fig. 7.

- (New)
1. To reproduce the resonant angle-resolved photo emission spectroscopy (R-ARPES) results, we set the parameters $\Delta_{\text{E1}} = 0.172$ eV and $\Delta_{\text{E2}} = 0.042$ eV in the effective tight-binding model (see each definition in the Methods). In addition, we set Δ_{ASO} and Δ_z as variable parameters so that the experimental λ_{IEE} can be reproduced by the calculation.
 2. From the band structure obtained in process 1, we theoretically calculated the Fermi level E_{F} dependence of the DOS and n_{sheet} . We measured the Hall effect in LTO (3 u.c.)/STO, from which we estimated n_{sheet} to be $\sim 8.9 \times 10^{13} \text{ cm}^{-2}$ at 4 K. We determined the Fermi level E'_{F} of the LSMO/LTO/STO sample from this n_{sheet} using the calculated E_{F} dependence of n_{sheet} (see Supplementary Fig. 9).
 3. We checked whether the theoretical λ_{IEE} using the E'_{F} value obtained above can reproduce the experimental λ_{IEE} or not. If not, we went back to process 1, changed the values of Δ_{ASO} and Δ_z , and followed the process 1 to 3 again.

After repeating the above-mentioned procedure, we found that the conditions of $\Delta_{\text{ASO}} = 0.015$ eV and $\Delta_z = 0.02$ eV can well reproduce the experimental λ_{IEE} . In Supplementary Fig. 9, one can see that $E_{\text{F}} = E'_{\text{F}} = -0.102$ eV corresponds to $n_{\text{sheet}} = \sim 8.9 \times 10^{13} \text{ cm}^{-2}$.

The differences in n_{sheet} and E_{F} of LTO/STO between the measurement methods, *i.e.* R-ARPES and the Hall measurement, may be due to the difference in oxidization of the sample. For the R-ARPES measurement, the oxidation was suppressed because the sample was transferred under a full nitrogen atmosphere. Meanwhile, for the Hall measurement, the sample was exposed to the atmosphere, and thus the surface was more oxidized. In LSMO/LTO/STO, the LTO layer is thought to be influenced by the large ozone flux during the growth of the LSMO layer. Thus, we think that the oxidation status of the LTO layer in LSMO/LTO/STO is closer to that of the LTO/STO sample just before the Hall measurement. Thus, we used n_{sheet} and E'_{F} estimated from the Hall effect for the calculation of λ_{IEE} .

- ✧ We changed the description about the Fermi surface on page 12 in the revised Supplementary Information.

Comment 3:

Nature of the transport in STO:

The authors state that they have a 2DEG but due to the low sheet resistance and high temperature deposition it is unclear if it is the case and the ARPES data are not very clear. The authors deposit LTO at a high temperature of 600°C and a low oxygen pressure of 2×10^{-7} Pa, such high temperature and low-pressure growth is known to induce oxygen vacancies as seen for example in Phys. Rev. Lett. **98**, 216803 (2007) or EPL **91**, 17004 (2010) for depositions at higher temperatures of 750°C and 800°C. These oxygen vacancies extend deeply in the bulk of STO (see Nature Mater. **7**, 621–625 (2008)) and lead to a low sheet resistance, high mobility and higher carrier density. Are the STO substrates still insulating after a similar treatment without deposition of the LTO?

Our response:

To check the status of oxygen vacancies, we annealed an STO substrate at a high temperature of 600 °C and a low oxygen pressure of 2×10^{-7} Pa, which are the conditions used for deposition of the $\text{LaTiO}_{3+\delta}$ layer in the LSMO/ $\text{LaTiO}_{3+\delta}$ /STO sample. We carried out the annealing for 10 min, which is longer than the growth time of the 3 u.c. $\text{LaTiO}_{3+\delta}$ layer. Figure R3 shows the temperature T dependence of R_{sheet} measured for the surface of this substrate. Here, we cut the annealed STO substrate into a small piece with a size of 4×2.35 mm and connected gold wires to the contacts at both edges of the sample (electrode distance is 1.6 mm). We measured the resistivity by a two-terminal method because using a four-terminal method is difficult due to the too high resistivity. We clearly see that the STO substrate is still insulating from room temperature to low temperature. Therefore, our growth condition does not affect the transport property of the STO substrate.

Fig. R3. Temperature (T) dependence of R_{sheet} for the STO substrate annealed at a high temperature of 600 °C and a low oxygen pressure of 2×10^{-7} Pa for 10 min.

Comment 4:

Do the authors have any experimental evidence of the two-dimensional nature of the transport in LTO/STO deposited using their technique?

Our response:

As explained below, the ARPES data shown in Fig.4a,b in the main manuscript show clear evidence of the quantization of the d bands, indicating the two-dimensional nature of electrons at the $\text{LaTiO}_{3+\delta}$ /STO interface. In Fig. 4a (in the previous manuscript), we can see that the bottom of the d_{xy} band is located at ~ -0.254 eV. In Fig. 4b (in the previous manuscript), we can clearly see that the edge of the Fermi surface of the d_{zx} band in the k_y direction is located at $\sim 0.6 \pi/a$ from the R-ARPES and calculation results. If the Ti $3d$ orbitals were not quantized and the bottoms of the d_{xy} , d_{yz} and d_{zx} bands degenerated at -0.254 eV as shown below in Fig. R4b, the Fermi surfaces of the d_{yz} and d_{zx} bands at the Fermi level ($= 0$ eV) would be much larger than that measured ($\sim 0.6 \pi/a$) [Santader-Syro, A. F. et al. *Nature* **469**, 189 (2011)]. Therefore, we can conclude that electrons are quantized (Fig. R4a in this response letter) and that the electronic channel at the $\text{LaTiO}_{3+\delta}$ /STO interface is two-dimensional.

We also measured the resistance between the top side of the $\text{LaTiO}_{3+\delta}$ layer and the back side of the STO substrate; the resistance was beyond 10 G Ω (= overrange). Therefore, we conclude that the conduction occurs not in the overall STO substrate but

near the $\text{LaTiO}_{3+\delta}/\text{STO}$ interface.

The values of the carrier density obtained for the $\text{LaTiO}_{3+\delta}/\text{STO}$ interface in our sample are typical for the 2DEG at STO interfaces. Other groups showed two-dimensional nature of carriers with n_{sheet} of $\sim 1.9 \times 10^{14} \text{ cm}^{-2}$ at the STO surface by ARPES measurements [Walker, S. M et al. *Adv. Mater.* **27**, 3894 (2015), Wang, Z et al. *Nat. Mater.* **15**, 837 (2016)]. Some other groups showed two-dimensional nature of the 2DEG with n_{sheet} of $\sim 4 \times 10^{14} \text{ cm}^{-2}$ at the GAO/STO interface using angle-resolved Shubnikov-de Haas oscillations [Chen, Y. Z. et al. *Nat. Commun.* **4**, 1371 (2013)]. As we noted in the response to comment 1, we estimated n_{sheet} at the $\text{LaTiO}_{3+\delta}/\text{STO}$ interface to be $\sim 8.9 \times 10^{13} \text{ cm}^{-2}$ by the Hall effect and to be $\sim 3 \times 10^{14} \text{ cm}^{-2}$ by the R-ARPES measurement using the theoretical calculation. These values are comparable to the above-mentioned values reported by other groups. Therefore, we can conclude that there are 2D carriers at the $\text{LaTiO}_{3+\delta}/\text{STO}$ interface in our sample.

Fig. R4. Band dispersions calculated by the effective tight-binding model. **a**, Calculated result *with* quantum confinement, which is the same as Fig. 4a of the previous manuscript. **b**, Calculated result *without* quantum confinement. The band dispersion shown in Fig. 4a of the previous manuscript and Fig. R4a can be obtained by adding terms of the confinement effect Δ_{E1} and Δ_{E2} , the spin-orbit interaction Δ_{ASO} and the polar lattice distortion Δ_z to the Hamiltonian used for obtaining the band dispersion shown in this figure (Fig. R4b).

Comment 5:

Metallicity of the LTO:

The authors state that the LTO they use is conductive due to the excess of oxygen.

Nonetheless they show no evidence of the metallicity of the LTO in this work. I understand it could be difficult in the LSMO/LTO/STO sample as LTO is only 3 uc but it could be simpler in a bilayer of LTO/STO with thick LTO where the contribution of LTO is dominant as shown in Phys. Rev. B 81, 161101 (2010).

In the supplementary information the authors present a sample with thick LTO (20 uc) on STO (figure S3), can they confirm that in this sample the LTO is metallic and its contribution to the transport is dominant?

Our response:

Following the reviewer’s suggestion, we conducted the transport measurements for the LaTiO_{3+δ} (20 u.c.) layer grown on an STO substrate. It shows metallic behavior as shown in Fig. R5 as is expected. Furthermore, we show the thickness t dependence of R_{sheet} for the LaTiO_{3+δ} layer (t nm) grown on an STO substrate (Fig. R6). R_{sheet} gradually decreases with increasing t . These results clearly indicate that LaTiO_{3+δ} is metallic. The decrease in R_{sheet} with increasing t is due to the increase in the LaTiO_{3+δ} thickness and the accompanying decrease in interface scattering as well as the increase in oxygen vacancies in the STO substrate. The metallicity of LaTiO_{3+δ} is also reported in many of previous works on LaTiO_{3+δ}/STO [Wong, F. J. et al. *Phys. Rev. B* **81**, 161101(R) (2010), Veit, M. J. et al. *Nat. Commun.* **9**, 1458 (2018), Edmondson, B. I. et al. *J. Appl. Phys.* **124**, 185303 (2018), Zhang, T. T. et al. *Appl. Phys. Lett.* **115**, 261604 (2019).].

Fig. R5. T dependence of the sheet resistance R_{sheet} for the LaTiO_{3+δ} (20 u.c.) layer grown on STO with a substrate temperature of 600 °C and a low oxygen pressure of 2×10^{-7} Pa.

Fig. R6. t dependence of R_{sheet} for the $\text{LaTiO}_{3+\delta}$ (t nm) layer grown on STO with a similar condition (substrate temperature of 720 °C with a low oxygen pressure of 2×10^{-7} Pa). The measurement temperature is 300 K.

➤ **Corresponding revised parts in the manuscript:**

- ✧ We added the thickness dependence of R_{sheet} for the $\text{LaTiO}_{3+\delta}$ (Fig. R6) on page 6 in the revised Supplementary Information.
- ✧ We added a following sentence in line 115–116 on page 5–6 in the revised manuscript:

(New) We can also confirm the metallicity of LTO made with our growth conditions in Supplementary Fig. 5.

Comment 6:

In reference 7 and Phys. Rev. B **81**, 161101 (2010) the authors associate the conductivity of LTO to the strain due to epitaxial deposition on STO while the authors associate it with oxygen excess. Can the authors explain why they consider the oxygen excess and not the strain?

Our response:

Dymkowski and Ederer predicted that about -2% strain (= compressive strain) closes the Mott gap and induces an insulator-to-metal transition in LaTiO_3 [Dymkowski, K. and Ederer, C. *Phys. Rev. B* **89**, 161109 I (2014), Sclauzero, G. et al. *Phys. Rev. B* **94**,

245109 (2016)]. On the other hand, F. Lichtenberg et al. showed that a small amount of excess oxygen induces metallic behavior in $\text{LaTiO}_{3+\delta}$ [Lichtenberg, F. et al. *Z. Phys. B* **82**, 211 (1991)]. Zhang et al. investigated the origin of the conductivity of LaTiO_3 by growing it on various substrates [Zhang, T. T. et al. *Appl. Phys. Lett.* **115**, 261604 (2019)]; they prepared LaTiO_3 films on $(\text{LaAlO}_3)_{0.3}(\text{Sr}_2\text{TaAlO}_6)_{0.7}$ (LSAT), STO and (110) TbScO_3 (TSO) substrates. The (compressive) strains applied to the epitaxial LaTiO_3 films on LSAT, STO and TSO substrates are -2.4% , -1.5% and -0.07% , respectively. Only the LaTiO_3 layer grown on STO showed metallic behavior, and $\text{LaTiO}_3/\text{LSAT}$ and $\text{LaTiO}_3/\text{TSO}$ showed insulating behavior. If the scenario of the compressive strain were correct, the $\text{LaTiO}_3/\text{LSAT}$ sample would show metallic behavior. They also showed the transport properties of $\text{LaTiO}_{3+\delta}$ grown on LSAT systematically changing the oxygen pressure during the growth. With increasing the oxygen pressure, the insulating behavior changed to metallic behavior in $\text{LaTiO}_{3+\delta}$ on LSAT. This is because $\text{LaTiO}_{3+\delta}$ grown on STO can easily absorb oxygen atoms from STO and the excess oxygen atoms generate hole carriers, partially changing the original Ti^{3+} ($3d^1$) states to the Ti^{4+} ($3d^0$) states. Therefore, Zhang et al. concluded that the metallicity of $\text{LaTiO}_{3+\delta}$ on STO is due to excess oxygen. We carried out similar experiments by using STO and LSAT substrates and confirmed the insulating behavior of $\text{LaTiO}_3/\text{LSAT}$ (not shown). Therefore, we think that the conductivity of $\text{LaTiO}_{3+\delta}$ originates from excess oxygen rather than strain.

Comment 7:

Role of the metallicity of the LTO:

The authors observe a very high conversion efficiency and associate it with the metallicity of the LTO layer. They write: “The large λ_{IEE} values obtained in our study suggest that the metallicity and the correlated transport with a large Coulomb repulsion of LTO likely play significant roles in spin current transport and in the enhancement of the spin-current conversion efficiency.” But in the case of the inverse Rashba Edelstein effect, the spin to charge conversion should be higher with an insulating interface as shown for example by Fert and Zhang in *Phys. Rev. B* 94, 184423 (2019). Indeed, as stated by the authors the conversion efficiency is proportional to the momentum relaxation time of the electrons τ_e , with $\lambda_{\text{IEE}} = \alpha_R * \tau_e / \hbar$, and a conductive interface would reduce the momentum relaxation time as an additional relaxation channel.

Can the authors comment on that apparent discrepancy between previous works, for spin to charge conversion an insulating interface is favorable, and their statement on conductive interfaces?

Our response:

As the reviewer pointed out, the momentum relaxation time is important for the inverse Edelstein effect (IEE); however, the decay of the spin current through the middle layer between the 2DEG and the ferromagnetic layer also reduces the IEE efficiency. The spin-to-charge conversion efficiency λ_{IEE} is determined by the balance between these two factors.

λ_{IEE} is determined by the Rashba coefficient α_{R} and relaxation time τ_e as seen in the equation $\lambda_{\text{IEE}} = \alpha_{\text{R}} \tau_e / \hbar$, where \hbar is the Dirac constant. On the other hand, the spin current j_{S} is also one of the most important parameters for λ_{IEE} as seen in the equation $\lambda_{\text{IEE}} = j_{\text{C}}^{2\text{D}} / j_{\text{S}}$. Here, $j_{\text{C}}^{2\text{D}}$ and j_{S} are the two-dimensional converted charge current and spin current, respectively. j_{S} is estimated at the interface between the ferromagnetic layer (FM) and the nonmagnetic layer (NM). When NM is insulating, there is a large loss for j_{S} when it passes through the NM layer before reaching the NM/STO interface [Song, Q. et al. *Sci. Adv.* **3**, 31602312 (2017), Wang, Y. et al. *Nano Lett.* **17**, 7659 (2017)]. The large loss of the spin current in diamagnetic insulators, such as STO, LAO and AlO_x , has been experimentally demonstrated by spin pumping measurements [Wang, H. et al. *Phys. Rev. B* **91**, 220410 (R) (2015)]. In other words, as the reviewer pointed out [Zhang, S. & Fert, A. *Phys. Rev. B* **94**, 184423 (2016)], the 2DEG at the STO interface has long τ_e when using insulating layers as a middle layer because the converted charge current only flows in the 2DEG at the STO interface, but insulating layers cause a loss of j_{S} .

In our work, we found that $\text{LaTiO}_{3+\delta}$ is an appropriate material in this point of view. $\text{LaTiO}_{3+\delta}$ has a good spin propagation because of its metallicity. Nevertheless, the resistivity ρ of $\text{LaTiO}_{3+\delta}$ is *not so low*; ρ is only about $1 \times 10^{-3} \Omega \cdot \text{cm}$ [Zhang, T. T. et al. *Appl. Phys. Lett.* **115**, 261604 (2019)], which is much higher than that of ordinary metals. We think that this “moderate” feature of $\text{LaTiO}_{3+\delta}$ can prevent the reduction of the effective momentum relaxation time of 2D electrons. Actually, LSMO/ $\text{LaTiO}_{3+\delta}$ shows the sheet resistance R_{sheet} of $\sim 200 \Omega/\square$ (on the LSAT substrate), which is not so low. R_{sheet} of $\sim 2 \Omega/\square$ of the 2DEG at the $\text{LaTiO}_{3+\delta}$ /STO interface is two orders of magnitude smaller than that of LSMO and $\text{LaTiO}_{3+\delta}$ as shown in Fig. 2a of the main manuscript. Therefore, in our LSMO/ $\text{LaTiO}_{3+\delta}$ /STO sample, most of the current flows at the $\text{LaTiO}_{3+\delta}$ /STO interface so that the relaxation time is not reduced significantly.

In addition, the sharpness of the interface in our heterostructures is also thought to significantly increase the relaxation time due to the suppression of interface roughness scattering. We have successfully realized this high-quality interface by MBE, which is

confirmed in the scanning transmission microscopy image (Fig. 1e in the main manuscript).

Comment 8:

Role of the antiferromagnetism in LTO:

The authors state that “antiferromagnets generally have excellent spin-current propagation, the inherent property of the antiferromagnetism of the Mott insulator LTO may be somewhat related to efficient spin injection into the 2DEG through LTO.” It is indeed true that insulating antiferromagnets, have excellent spin-current propagation properties, for example NiO. especially close to the Néel temperature as seen in Phys. Rev. Lett. 116, 186601 (2016).

Can the author comment on the Néel temperature of LTO? Is it compatible with highly efficient spin current injection?

Our response:

We would like to thank the reviewer for the kind comment. LaTiO_3 is an antiferromagnetic Mott insulator, and the Néel temperature is ~ 150 K [Lichtenberg, F. et al. *Z. Phys. B* **82**, 211 (1991)]. λ_{IEE} largely increases from ~ 150 K as shown in Fig. 3e in the main manuscript. Although it is not clear at this time, it is possible that a small amount of antiferromagnetic region remains in paramagnetic metal $\text{LaTiO}_{3+\delta}$, which may increase the IEE.

➤ **Corresponding revised parts in the manuscript:**

✧ We added a description about the Néel temperature in line 181–183 on page 8 in the revised manuscript:

(Old) In addition, considering that antiferromagnets generally have excellent spin-current propagation³⁰, the inherent property of the antiferromagnetism of the Mott insulator LTO may be somewhat related to efficient spin injection into the 2DEG through LTO.

(New) In addition, considering that antiferromagnets generally have excellent spin-current propagation³⁸, the inherent property of the antiferromagnetism of the Mott insulator LTO (Néel temperature is ~ 150 K)¹⁹ may be somewhat related to the efficient spin injection into the 2DEG through LTO possibly due to the existence of small amount of the antiferromagnetic region.

Comment 9:

Comments on the ARPES data:

I am not a specialist of ARPES but it seems that the quality of the ARPES data is not good enough to clearly observe the bandstructure of LTO/STO and the calculated bandstructure doesn't look like the experimental data. It is especially unclear to me how it is possible to observe the bandstructure of 2DEG in STO without any additional contribution of the LTO if it is conductive (states at the fermi level in LTO should appear).

Our response:

Generally, it is extremely difficult to observe electronic structures of “buried in heterostructures” like our case. We have achieved it using resonant ARPES using a soft x-ray (SX) light, which is best suited for studying the band structure of interfaces. This is a really challenging experiment. [We note that other groups have reported the “surface” band structure of STO using a vacuum ultraviolet (VUV) light. This is much easier, but this does not necessarily capture the real band structure of buried interfaces.] ARPES using an SX light can probe the signal of 2D conductivity related to the Ti states at the buried $\text{LaTiO}_{3+\delta}$ /STO interface with enough penetration depth and sufficient resonant enhancement by selecting the photon energy at the Ti L_3 edge ($h\nu = 459.7$ eV). The spectra obtained in this study are very similar to the previously reported challenging SX-ARPES results for the buried heterointerfaces such as $\text{LaAlO}_3/\text{SrTiO}_3$ [Berner, G. et al. *Phys. Rev. Lett.* **110**, 247601 (2013), Cancellieri, C. et al. *Phys. Rev. B* **89**, 121412I (2014), Hong, H. et al. *Phys. Rev. Mater.* **6**, L011401 (2022)]. As explained in the caption of Fig. 4, part of the Fermi surface is missing in comparison with the calculated one due to the experimental geometry of the ARPES measurements including the polarization of the incident SX light [Cancellieri, C. et al., *Phys. Rev. B* **89**, 121412I (2014)]. Although the experimental band dispersion is likely unresolved due to the energy resolution with the SX light, the observed Fermi surface elongated along the k_y direction and the bottom of the band dispersion agree well with the calculated band structure.

The observed Fermi surface may partially include the band component of $\text{LaTiO}_{3+\delta}$. Although it is not clear at this time, the excess oxygen atoms in $\text{LaTiO}_{3+\delta}$ may generate a small Fermi surface with a relatively small density of hole carriers. Thus, the Fermi surface of $\text{LaTiO}_{3+\delta}$ may be overlapped at the center of the Fermi surface observed in our study, although it is hard to distinguish.

➤ Corresponding revised parts in the manuscript:

✧ We added the sentence about the overlap of the Fermi surface of the thin $\text{LaTiO}_{3+\delta}$

layer in line 514–516 on page 23 in the revised manuscript:

(New) Although it is not clear, the Fermi surface of the thin LTO layer, which is considered to be small due to its small carrier concentration, may be partially overlapped with the Fermi surface of the 2DEG at around the center of the k_x - k_y plane.

✧ We added the citation about the polarization of the incident SX light as ref. 41 in line 518 on page 23 in the revised manuscript:

(Old) In this measurement, the Fermi surface elongated in the k_y direction is visible, but the surface elongated in the k_x direction is almost not, due to the experimental geometry of ARPES measurements.

(New) In this measurement, the Fermi surface elongated in the k_y direction is visible, but the surface elongated in the k_x direction is almost not, due to the experimental geometry of ARPES measurements⁴¹.

Comment 10:

Can the authors comment on the differences between the ARPES and the calculated bandstructure?

Our response:

The differences originate from the usage of the linear horizontal light and horizontal detection slit geometry, which result in the observation of the electronic states in Fig. 4a derived mainly from the d_{zx} orbitals and slightly from d_{xy} and d_{yz} orbitals. In Fig. 4b, only the Fermi surface elongated in the k_y direction (*i.e.* d_{zx} bands) is visible, but the one elongated in the k_x direction (d_{yz} band) is almost invisible, due to the geometry of our ARPES measurements [Cancellieri, C. et al., *Phys. Rev. B* **89**, 121412I (2014)]. This is due to the dipole selection rules [Damascelli, A. et al. *Rev. Mod. Phys.* **75**, 473 (2003)]. Although the APRES spectra appear to differ from the band calculation results at a glance, the positions of the d_{xy} and d_{zx} bands can be properly discussed from the data of the observed ARPES spectra.

➤ **Corresponding revised parts in the manuscript:**

✧ For more precise fitting to the R-ARPES results, we changed the parameters Δ_{E1} and Δ_{E2} from 0.12 eV and 0.02 eV to 0.172 eV and 0.042 eV, respectively, for the calculation of the band structure, Fermi surface, Fermi energy, and theoretical λ_{IEE} in Fig. 4a,b,c,d of the revised manuscript and Supplementary Fig. 9 and 10.

Comment 11:

Can the authors comment on the absence of the d-states of conductive LTO in the ARPES? To sum up in its current form the manuscript is unclear on the exact electrical properties of LTO (insulating or conductive) and of the 2DEG properties. The role of the metallicity or antiferromagnetism of LTO also have to be clarified to better understand why such a high conversion efficiency is obtained.

Our response:

In summary, as mentioned above, the observed Fermi surface may partially include the band component of $\text{LaTiO}_{3+\delta}$. The excess oxygen atoms in $\text{LaTiO}_{3+\delta}$ may generate a small Fermi surface due to the small density of hole carriers. Thus, the Fermi surface of $\text{LaTiO}_{3+\delta}$ may be overlapped at the center of the Fermi surface observed in our study. As shown in Fig. R5 and R6, $\text{LaTiO}_{3+\delta}$ is clearly metallic, as evidenced by the temperature and thickness dependences of R_{sheet} . The 2DEG of our LSMO/ $\text{LaTiO}_{3+\delta}$ /STO heterostructure has $R_{\text{sheet}} = \sim 8.9 \times 10^{13} \text{ cm}^{-2}$ (by the Hall effect) and $n_{\text{sheet}} = \sim 8 \text{ } \Omega/\square$, which are typical values for the 2DEG on STO (light orange zone in Fig. R1). Concerning the antiferromagnetism, the antiferromagnetic region may partially remain in $\text{LaTiO}_{3+\delta}$, and it may increase the IEE. The Néel temperature is $\sim 150 \text{ K}$ [Lichtenberg, F. et al. *Z. Phys. B* **82**, 211 (1991)]. The metallicity of $\text{LaTiO}_{3+\delta}$ is important for the high spin-to-charge conversion efficiency. Nevertheless, the resistivity ρ of $\text{LaTiO}_{3+\delta}$ is *not so low*; ρ is only about $1 \times 10^{-3} \text{ } \Omega \cdot \text{cm}$ [Zhang, T. T. et al. *Appl. Phys. Lett.* **115**, 261604 (2019)], which is much higher than ordinary metals, preventing the reduction of the effective momentum relaxation time of the 2D electrons.

Comment 12:**Other comments:**

-The authors want to have LTO with oxygen in excess but deposit LTO at very low oxygen pressure, how is it compatible?

Our response:

LaTiO_3 is an oxygen-sensitive metastable phase. In general, no matter how low the oxygen pressure is used for the growth of LaTiO_3 on STO, LaTiO_3 becomes overoxidized due to easy diffusion of oxygen atoms from STO to LaTiO_3 [Zhang, T. T. et al. *Appl. Phys. Lett.* **115**, 261604 (2019), Edmondson, B. I. et al. *J. Appl. Phys.* **124**,

185303 (2018), Scheiderer, P. et al. *Adv. Mater.* **30**, 1706708 (2018), Lin, C. et al. *Phys. Rev. B* **92**, 035110 (2015)]. Therefore, $\text{LaTiO}_{3+\delta}$ tends to be overoxidized on STO.

➤ **Corresponding revised parts in the manuscript:**

✧ We added a description about the over-oxidization of LTO grown on an STO substrate in line 95–96 on page 5 in the revised manuscript:

(New) We note that LTO grown on STO is overoxidized due to the easy diffusion of oxygen atoms from STO to LTO^{20,23,27,28}.

Comment 13:

-In the methods section it is unclear if the targets used are metallic or oxides, can the author clarify this point. If metallic can the author comment on the possible role of Ti in the formation of oxygen vacancies?

Our response:

We used pure metallic La, Ti, Sr and Mn sources filled in Knudsen cells in MBE, which is a standard way for MBE growth. We added the explanation of this point in the revised manuscript. As explained above, as for $\text{LaTiO}_{3+\delta}$, no oxygen vacancy is thought to exist. Meanwhile, oxygen vacancies are generated in STO during the growth of $\text{LaTiO}_{3+\delta}$.

➤ **Corresponding revised parts in the manuscript:**

✧ We added a description about our material sources in line 235–236 on page 10 in the revised manuscript:

(Old) We used a shuttered growth technique with La, Ti, Sr and Mn fluxes supplied by Knudsen cells.

(New) We used a shuttered growth technique with fluxes supplied from pure metallic La, Ti, Sr and Mn sources in Knudsen cells.

Comment 14:

-The authors do not show any data on the damping versus the temperature contrary to their previous work. It would be interesting to have these data as well as the damping versus temperature in their reference sample for comparison purposes. The estimated spin current injected and spin mixing conductance should also be shown.

Our response:

Here, we show the temperature T dependence of the damping constant α , mixing conductance $g_r^{\uparrow\downarrow}$ and spin current j_s (Fig. R7). We added the data in the revised Supplementary Information.

Fig. R7. T dependence of **a**, α , **b**, $g_r^{\uparrow\downarrow}$, **c**, j_s .

➤ Corresponding revised parts in the manuscript:

- ✧ We added the data of the α , $g_r^{\uparrow\downarrow}$ and j_s (Fig. R7) on page 8 in the revised Supplementary Information.
- ✧ We added a following sentence in line 272–273 on page 12 in the revised manuscript:
(New) The obtained values of α , $g_r^{\uparrow\downarrow}$, and j_s^0 are shown in Supplementary Fig. 7.

Comment 15:

-The authors use LSMO/LTO/LSAT as their reference sample. As LSAT has very different dielectric properties compared with STO I am not sure that is the ideal reference sample. Due to its large dielectric constant and dielectric loss STO can slightly affect the cavity properties and lead to enhanced rectification effects. Do the authors have similar spin pumping FMR measurements in LSMO/STO?

Our response:

Yes, we conducted the spin pumping measurements for LSMO/STO. As shown in Fig. R8, in comparison with the result obtained for LSMO/LaTiO_{3+ δ} /STO, the converted charge current is negligibly small in LSMO/STO (see the green curve). We added this data in the revised manuscript.

Fig. R8. Comparison of j_c^{2D}/h_{rf}^2 for LSMO/LaTiO_{3+δ}/STO, LSMO/LaTiO_{3+δ}/LSAT and LSMO/STO measured with the microwave power MP of 50 mW. The measurements are conducted at 15 K.

➤ **Corresponding revised parts in the manuscript:**

- ✧ We added the data of the converted charge current measured for LSMO/STO in Fig. 3d of the revised manuscript.
- ✧ We added descriptions about the LSMO/STO sample in line 100–108 on page 5 in the revised manuscript:

(Old) In addition, we grow a reference sample comprising LSMO (30 u.c.)/LTO (3 u.c.) on an (LaAlO₃)_{0.3}(Sr₂TaAlO₆)_{0.7} (LSAT) (001) substrate under the same conditions. This reference sample does not have a 2DEG, so we can use it to check whether the LTO layer itself has any influence on the IEE. The LTO layer in both the LSMO/LTO/STO and LSMO/LTO/LSAT samples becomes metallic due to the high oxygen pressure used for growing LSMO¹³.

(New) In addition, we grow **reference samples** comprising LSMO (30 u.c.)/LTO (3 u.c.) on an (LaAlO₃)_{0.3}(Sr₂TaAlO₆)_{0.7} (LSAT) (001) substrate **and LSMO (30 u.c.) on an STO (001) substrate** under the same **growth conditions as those used for the LSMO/LTO/STO sample**. The LSMO/LTO/LSAT sample does not have a 2DEG, so we can use it to check whether the LTO layer itself has any influence on the IEE. The LTO layer in both the LSMO/LTO/STO and LSMO/LTO/LSAT samples becomes metallic due to the high oxygen pressure used for growing LSMO²⁰. **Using the reference LSMO/STO sample, we can eliminate the possible influence of the rectification effects of LSMO in the spin pumping measurements.**

✧ We added the results of the spin pumping measurement for the LSMO/STO sample in line 154–155 on page 7 in the revised manuscript:

(New) The $j_{C,0^\circ}^{2D}/h_{\text{rf}}^2$ obtained for LSMO/STO is also almost zero, showing no influence of the rectification effect of LSMO.

✧ We added a description of the LSMO/STO sample in line 496–497 on page 22 in the revised manuscript:

(Old) Comparison of j_C^{2D}/h_{rf}^2 between LSMO/LTO/STO and LSMO/LTO/LSAT measured with $MP = 50$ mW.

(New) Comparison of $j_{C,0^\circ}^{2D}/h_{\text{rf}}^2$ between LSMO/LTO/STO, LSMO/LTO/LSAT and LSMO/STO measured with $MP = 50$ mW.

Comment 16:

-The authors write “To extract a pure IEE signal, we decompose the EMF into a symmetric (Lorentzian) component V_{sym} , which includes signals of the IEE, and an antisymmetric (anti-Lorentzian) component V_{asym} .” But this is generally not true the rectification effects can have a symmetric and an antisymmetric component as shown in *Phys. Rev. B* 88, 064403 (2013) for example. Ideally an angular dependence measurement should be performed. As the author observed no effect in a similar sample without a 2DEG the contribution of the rectification effects should be small.

Our response:

As the reviewer pointed out, symmetric voltage V_{sym} includes rectification effects, such as planer Hall effect (PHE) and anomalous Hall effect (AHE). One can deny these possibilities by measuring the angular dependence of the electromotive force (EMF) [Rojas-Sánchez, J. -C. et al. *Phys. Rev. B* 88, 064403 (2013)], as mentioned by the reviewer, or by the T dependence of the EMF [Ohya, S. et al. *Phys. Rev. B* 96, 094424 (2017)]. The PHE and AHE are known to be proportional to ρ^n ($n = 1-2$, depending on the scattering mechanism), where ρ is the resistivity. Since ρ decreases significantly with decreasing T in LSMO, the PHE and AHE contributions also become very small with decreasing T . Meanwhile, the EMF obtained by spin pumping increases with decreasing T (see the inset of Fig. 3c in the revised manuscript), which means that the EMF can be attributed to the spin-to-charge conversion at the LaTiO_{3+δ}/STO interface rather than to the PHE and AHE in LSMO.

Furthermore, as already pointed out by the reviewer, we see only a negligibly small spin-to-charge conversion signal in LSMO/STO and LSMO/LaTiO_{3+δ}/LSAT in Fig.

R8, eliminating the influence of the PHE and AHE of LSMO on our signal for LSMO/LaTiO_{3+δ}/STO.

Comment 17:

-In figure 4c) when the spin orbit interaction decreases the conversion efficiency increases, this is very counter intuitive can the authors comment on this?

Our response:

λ_{IEE} in the 2DEG at the STO interface is not necessarily proportional to the magnitude of the atomic spin-orbit interaction Δ_{ASO} . This is because the signs of the spin splitting are different among the d_{xy} , d_{yz} and d_{zx} bands and because spin splitting is larger in the bands closer to band crossing points in the k_x - k_y plane. Thus, the sign and magnitude of the spin splitting are different depending on the k point. Which spin splitting is dominant changes depending on the Fermi level position. Therefore, stronger spin-orbit interaction does not necessarily lead to larger λ_{IEE} .

However, the splitting width Δ for “each” band (in the k_x direction) is proportional to Δ_{ASO} . At $E_F = 0$ meV (see the Fermi surface in Supplementary Fig. 8f,k of the previous supplementary information), at which the Fermi surface is large enough to neglect the influence of the band crossings, λ_{IEE} is proportional to the magnitude of the spin-orbit interaction. This is because Δ of the d_{yz} and d_{zx} bands, which are dominant over the d_{xy} band at $E_F = 0$ meV, is positive (see $E_F = 0$ meV in Fig. R9). Meanwhile, at $E'_F = -137$ meV, which was used for the calculation of λ_{IEE} (in the previous manuscript) and is located near the Lifshitz point, λ_{IEE} is not simply proportional to the magnitude of the spin-orbit interaction due to the complex d orbitals (see around the crossing point of the dotted lines in Fig. R9). In Supplementary Fig. 8d,i of the previous Supplementary information, the sign of Δ of the most dominant innermost band is different from those of other bands so that λ_{IEE} is decreased.

Fig. R9. E_F dependence of calculated λ_{IEE} for $\Delta_{\text{ASO}} = 10, 15$ and 20 meV. The crossing point of the dotted lines is the λ_{IEE} and estimated E_F ($= E'_F = -0.137$ eV, which was used in the previous manuscript) in LSMO/LaTiO $_{3+\delta}$ /STO used for the spin pumping measurements.

Comment 18:

-In the introduction the authors wrote “Moreover, the mechanism of the spin-current flow in these insulating layers is still unexplored.” I would like to point out that a recent article studies this mechanism *Phys. Rev. Research* **3**, 043170 (2021).

Our response:

We would like to thank the reviewer for introducing the interesting paper [To, D. Q. et al., *Phys. Rev. Res.* **3**, 043170 (2021)], which shows that resonant tunneling leads to the gate-voltage dependent asymmetric tunneling and inverse Edelstein effect by a theoretical calculation. This idea really helps us understand the inverse Edelstein effect. We cite this paper for the mechanism of the IEE.

➤ **Corresponding revised parts in the manuscript:**

✧ We added this paper as ref. 7 in line 44 and 54 on page 3 in the revised manuscript:

(Old) The IEE has the great advantage of being artificially designed and controlled by creating interfaces that combine various materials^{2,3,4,5,9,10,11}.

(New) The IEE has the great advantage of being artificially designed and controlled by creating interfaces that combine various material²⁻⁹.

- (Old) Among these, the 2DEG formed at interfaces between perovskite-oxide STO and other oxides, such as LaAlO_3 (LAO)¹ and AlO_x , is capable of extremely efficient spin-to-charge current conversion^{2,3,4,5}.
- (New) Among these, the 2DEG formed at interfaces between perovskite-oxide STO and other oxides, such as LaAlO_3 (LAO)¹⁰ and AlO_x ¹¹, is capable of extremely efficient spin-to-charge current conversion³⁻⁷.

Reviewer #2's comments

In this work by Kaneta-Takada et al, the authors show a large spin pumping voltage signal in a 2DEG system based on the over oxygenated polar metal LaTiO_3 on SrTiO_3 substrate. As an injector layer, they use 12 nm of the LSMO manganite, which is also epitaxial. The voltage reaches values as high as 500 μV at 15 K! As the sample is poorly resistive at 15 K, due to the high conductivity of the 2D gas and the metallic character of the doped LaTiO_3 , the estimated efficiency turns out to be giant. The normalized value of the two-dimensional charge current production exceeds 600 mA/mG^2 , and the efficiency reaches values of 190 nm. A record. In addition, the authors show ARPES measurements with theoretical fit highlighting the nature of the two-dimensional gas. This result shows a new system very attractive to the community and opens new possibilities for further studies of spin-charge interconversion as well as potential applications. One would like to know for example about the anisotropy, or not, of the interconversion, as well as a dependence on the gate voltage. Surely that will be done in future work. The measurements are convincing, indeed the authors show a very attractive new system, with record values of spin current to charge current conversion. Therefore, my advice is to recommend the publication of this manuscript.

Our response:

We would like to really thank the reviewer for the positive and encouraging comments.

Comment 1:

The reviewer has some minor details to point out:

In the abstract, please add that the demonstration of giant spin-to-charge current conversions efficiencies is by Spin pumping FMR voltage measurements.

Our response:

We agree with this comment, which will help readers understand our manuscript more clearly. Thus, we added the explanation in the abstract.

➤ **Corresponding revised parts in the manuscript:**

✧ We added the sentence in line 32 on page 2 in the revised manuscript:

(Old) we demonstrate giant spin-to-charge current conversion efficiencies, up to ~190 nm.

(New) we demonstrate giant spin-to-charge current conversion efficiencies, up to ~190 nm, using spin-pumping ferromagnetic-resonance voltage measurements.

Comment 2:

Line 73-75: if there is a spin texture, such as spin-momentum locking, and which is schematized in Figure 4d, the relaxation time indicated in the equation on line 73 is not of the electron momentum, or of the spin relaxation time, but of both.

Our response:

We thank the reviewer for the insightful comment. We corrected the wording.

➤ **Corresponding revised part in the manuscript:**

✧ We changed the wording in line 75 on page 4 in the revised manuscript:

(Old) τ_e is the momentum relaxation time of electrons

(New) τ_e is the momentum/spin relaxation time of electrons

Comment 3:

How did the authors measure or estimate h_{rf} ? h_{rf} is used in equation (2). I have read reference (3) from the same group but have not found information on how the authors estimate h_{rf} . This parameter, which depends on the experimental setup, is important. A small error in h_{rf} will strongly impact the estimation of J_s and consequently the interconversion efficiency. Can the authors also indicate the model of their spectrometer? If the authors use h_{rf} given by the fabricator, that would be for an empty cavity. But when the sample is inserted, it changes its quality factor Q and thus the h_{rf} . Do the authors consider that h_{rf} does not change when the sample is inserted into the resonant cavity?

Our response:

As the reviewer pointed out, the microwave magnetic field h_{rf} is an important

parameter for the estimation of the spin current density j_s . We used the Electron Spin Resonance System (ESR) JES-FA300 made by JEOL. JEOL provided us the data of h_{rf} vs. quality factor Q . Before measuring each sample, we checked the Q value with the sample inserted into the cavity and estimated the h_{rf} value using this h_{rf} - Q data.

➤ **Corresponding revised parts in the manuscript:**

✧ We added the model name of the spectrometer and the explanation for the estimation of h_{rf} in line 252 and 273–275 on page 11–12 in the revised manuscript:

(Old) We carried out spin pumping measurements using a transverse electric (TE₀₁₁) cavity of an electron-spin resonance system with a microwave frequency of 9.1 GHz.

(New) We carried out spin pumping measurements using a transverse electric (TE₀₁₁) cavity of an electron-spin resonance system with a microwave frequency of 9.1 GHz in an electron spin resonance system JES-FA300, JEOL.

(New) Before measuring each sample, we checked the quality factor Q with the sample inserted into the cavity and estimated the h_{rf} value using the data of h_{rf} vs. Q provided by JEOL.

Comment 4:

I suppose that the measured voltage curves, electromotive force EMF in the authors' notation, are the raw data, right? It seems so from looking at the curves in Figure 3b and 3d. However, if so the authors should state it explicitly, or the authors show only the symmetrical contribution?

Our response:

The microwave absorption derivative, the electromotive force (EMF), and the two-dimensional charge current per microwave magnetic field square j_C^{2D}/h_{rf}^2 in Fig. 3 are raw data. To clarify this point, we defined the raw EMF, the raw EMF measured at $\theta_H = 0^\circ$, and the raw two-dimensional charge current obtained at $\theta_H = 0^\circ$ as V , V_{0° , and $j_{C,0^\circ}^{2D}$, respectively. We modified the descriptions of the vertical axis of Fig. 3b,c,d.

➤ **Corresponding revised parts in the manuscript:**

✧ We replaced the definition about the raw data on page 6–7 in the revised manuscript:

✧ We added a comment about the raw data on page 22 in the revised manuscript:

(Old) **b**, Magnetic field $\mu_0 H$ dependencies of the EMF at 15 K measured for the LSMO/LTO/STO sample with various MP values ranging from 30 to 100 mW. In the

electron-spin resonance system, a microwave magnetic field h_{mw} is applied along the $[1\bar{1}0]$ direction of the STO substrate. The inset shows the linear relation between the microwave power MP and V_{sym} . **c**, H dependence of j_C^{2D}/h_{rf}^2 measured for the LSMO/LTO/STO sample at various temperatures ranging from 15 to 300 K with $MP = 50$ mW. The inset shows the H dependence of the EMF. **d**, Comparison of j_C^{2D}/h_{rf}^2 between LSMO/LTO/STO and LSMO/LTO/LSAT measured with $MP = 50$ mW. The measurements are conducted at 15 K. The inset shows an enlarged figure of the $\mu_0 H$ dependence of j_C^{2D}/h_{rf}^2 measured for the LSMO/LTO/LSAT sample. **e**, Summary of the temperature dependences of λ_{IEE} in various material systems.

(New) **b**, Magnetic field $\mu_0 H$ dependencies of V at 15 K measured for the LSMO/LTO/STO sample with various MP values ranging from 30 to 100 mW. In the electron-spin resonance system, a microwave magnetic field h_{mw} is applied along the $[1\bar{1}0]$ direction of the STO substrate. The inset shows the linear relation between the microwave power MP and V_{sym} . **c**, H dependence of $j_{C,0^\circ}^{2D}/h_{rf}^2$ measured for the LSMO/LTO/STO sample at various temperatures ranging from 15 to 300 K with $MP = 50$ mW. The inset shows the H dependence of V_{0° . **d**, Comparison of $j_{C,0^\circ}^{2D}/h_{rf}^2$ between LSMO/LTO/STO, LSMO/LTO/LSAT and LSMO/STO measured with $MP = 50$ mW. The measurements are conducted at 15 K. The inset shows an enlarged figure of the $\mu_0 H$ dependence of $j_{C,0^\circ}^{2D}/h_{rf}^2$ measured for the LSMO/LTO/LSAT sample. **e**, Summary of the temperature dependences of $\lambda_{IEE}(=j_{C,sym}^{2D}/j_S^0)$ in various material systems.

Comment 5:

In line 26, it appears that the authors made a mistake or it is a typo. The 2D charge current is the measured spin pumping voltage normalized by the total sample resistance and the bar width (0.95 mm), not the distance between the electrodes (1.2 mm) as stated in the manuscript. Please revise that

Our response:

The equation described in our manuscript is correct; the 2D charge current j_C^{2D} is expressed as $j_C^{2D} = V_{sym}/R_{sheet}l = V_{sym}/Rw$, where V_{sym} , R_{sheet} , R , l and w are the symmetric voltage, sheet resistance, total resistance, length and width of the sample, respectively. Here, the relationship between R_{sheet} and R is expressed by $R_{sheet} = R \times w/l$. In summary, j_C^{2D} is expressed as below:

$$j_C^{2D} = \frac{V_{sym,ave}}{Rw} = \frac{V_{sym,ave}}{\frac{R_{sheet}l}{w}} = \frac{V_{sym,ave}}{R_{sheet}l}$$

Comment 6:

In the present work, the authors perform their measurements when H is applied along the easy axis of in-plane magnetic anisotropy, [110], and the voltage is measured at 90 degrees from that direction. Have the authors measured when H is applied in another direction? Could they comment on this? Have they chosen the easy axis because the resonance field would be lower, or because the efficiency is higher in this direction?

Our response:

We have chosen the easy axis of in-plane magnetic anisotropy in (La,Sr)MnO₃, which corresponds to the [110] direction of SrTiO₃, because the resonance field is expected to be the lowest in this direction. Another important reason for using this direction is to suppress the contribution of the planer Hall effect in the spin pumping signal; it becomes negligibly small when the magnetization direction is aligned along the magnetic field direction under the ferromagnetic resonance (FMR) condition in theory [Ohya, S. et al. *Phys. Rev. B* **96**, 094424 (2017).]. This is vital for the accurate estimation of the λ_{IEE} . We commented on this point in our revised manuscript.

➤ Corresponding revised part in the manuscript:

✧ We added the comment on the magnetic field direction in line 257 on page 11 in the revised manuscript:

(Old) which corresponds to the easy magnetization axis of LSMO (see ref. 3 and Supplementary Fig. 5).

(New) which corresponds to the easy magnetization axis of LSMO (see ref. 4 and Supplementary Fig. 6) **and can suppress the contribution of the planer Hall effect³³**.

Other minor revised points

We would like to revise ref. 42 because it has been published recently.

➤ Corresponding revised part in the manuscript:

✧ We revised the description of ref. 13 in line 356 on page 15 in the revised manuscript:

(Old) 42. Arai, S., Kaneta-Takada, S., Anh, L. D., Tanaka, M. & Ohya, S. Theoretical analysis of the inverse Edelstein effect at the LaAlO₃/SrTiO₃ interface with an effective tight-binding model: Important role of the second d_{xy} subband. *arXiv:2111.12275* (2021).

- (New) 13. Arai, S., Kaneta-Takada, S., Anh, L. D., Tanaka, M. & Ohya, S. Theoretical analysis of the inverse Edelstein effect at the LaAlO₃/SrTiO₃ interface with an effective tight-binding model: Important role of the second d_{xy} subband. *Appl. Phys. Express* **15**, 013005 (2022).

We would like to revise Supplementary Fig. 1 because of more accurate understanding.

➤ **Corresponding revised part in the manuscript:**

- ✧ We modified the Fermi level position in the Supplementary Fig. 1.

REVIEWERS' COMMENTS

Reviewer #1 (Remarks to the Author):

I would like to thank the authors for their very complete answers to my comments.

My concerns have been carefully addressed in their answers and the revised manuscript. I mentioned in my previous review that "in its current form, the manuscript is unclear on the exact electrical properties of LTO (insulating or conductive) and of the 2DEG properties. The role of the metallicity or antiferromagnetism of LTO also have to be clarified to better understand why such a high conversion efficiency is obtained".

The authors gave a very satisfying and complete answer on all these key points. As they modified the manuscript accordingly, and as the manuscript itself shows noteworthy results with record high conversion efficiency, I think it can be accepted to be published in Nature Communications.

Note that I would still appreciate seeing some very short discussion in the final version of the manuscript on the following two points:

-The authors used a 2DEG with a lower sheet resistance than other group studying the interconversion. As the mobility/relaxation time is a key parameter to obtain a large conversion efficiency, I think it is worth mentioning it and comparing the sheet resistance to others.

-I think the discussion of the authors on my comment number 7 on why a poorly metallic interface could be better than a semiconducting interface should appear somewhere in the manuscript. As it might appear to be a key ingredient to optimize the spin to charge conversion efficiency, and could be more straightforwardly compatible with electrical injection of spin current avoiding the high resistance of a tunnel barrier.

Reviewer #2 (Remarks to the Author):

The authors have responded satisfactorily to the comments of the two reviewers. This reviewer recommends the present manuscript for publication in Nature Communication. The reviewer highlights that this new system shows a record spin-charge current conversion efficiency, reaching raw data of 400 μV dc spin pumping voltage for a microwave power input of 100 mW, and an IEE efficiency value of almost 200 nm. This work sets another precedent for the spintronics and oxide community for instance.

Response letter

We would like to thank the reviewers for their positive and valuable comments on our manuscript (NCOMMS-22-00119A). We have carefully revised our manuscript following those comments. In this Response letter, we reply to all the comments, point by point, and describe how we revised the manuscript. Here, the comments of the reviewers are written in blue color. The revised parts are written in red color in this Response letter and the main manuscript.

Reviewer #1's comments

I would like to thank the authors for their very complete answers to my comments.

My concerns have been carefully addressed in their answers and the revised manuscript. I mentioned in my previous review that "in its current form, the manuscript is unclear on the exact electrical properties of LTO (insulating or conductive) and of the 2DEG properties. The role of the metallicity or antiferromagnetism of LTO also have to be clarified to better understand why such a high conversion efficiency is obtained".

The authors gave a very satisfying and complete answer on all these key points. As they modified the manuscript accordingly, and as the manuscript itself shows noteworthy results with record high conversion efficiency, I think it can be accepted to be published in Nature Communications.

Note that I would still appreciate seeing some very short discussion in the final version of the manuscript on the following two points:

Comment 1:

-The authors used a two-dimensional electron gas (2DEG) with a lower sheet resistance than other group studying the interconversion. As the mobility/relaxation time is a key parameter to obtain a large conversion efficiency, I think it is worth mentioning it and comparing the sheet resistance to others.

Our response:

We would like to thank the reviewer for the positive evaluation of our manuscript and the valuable comment. Following the reviewer's suggestion, we added the the mobility value of our $\text{LaTiO}_{3+\delta}$ (LTO) [3 unit-cell (u.c.)]/ SrTiO_3 (STO) heterostructure ($6.9 \times 10^3 \text{ cm}^2/\text{Vs}$) obtained by the Hall effect in the Supplementary Information. The

relaxation time is shown in Supplementary Fig. 8. We also added the discussion about the sheet resistance of our LTO/STO film in comparison with other interface systems as follows.

➤ **Corresponding revised parts in the manuscript:**

- ✧ We added the value of the mobility of LTO/STO and the following discussion about the sheet resistance of our (La,Sr)MnO₃/LTO/STO film on page 10–11 in the revised Supplementary information.

(New) As we mentioned above, we measured the Hall effect for the LTO (3 u.c.)/STO sample, for which the sheet carrier density n_{sheet} , mobility and the sheet resistance R_{sheet} were estimated to be $\sim 8.9 \times 10^{13} \text{ cm}^{-2}$, $6.9 \times 10^3 \text{ cm}^2/\text{Vs}$ and $\sim 8 \text{ } \Omega/\square$ at 4 K, respectively. R_{sheet} of $\sim 2.4 \text{ } \Omega/\square$ obtained for the LSMO/LTO/STO sample at 5 K (see Fig. 2a in the main manuscript, and the possible reason for the difference in R_{sheet} between LSMO/LTO/STO and LTO/STO is noted in Supplementary Note 4) is relatively low compared with other two-dimensional electron gas (2DEG) systems formed at STO interfaces. However, this R_{sheet} value is well within the range of the reported values as shown in Fig. R10, in which we summarize R_{sheet} vs. n_{sheet} reported for 2DEGs at various STO interfaces grown by sputtering, pulse laser deposition (PLD) and molecular beam epitaxy (MBE). As indicated by the green points obtained for LaAlO₃ (LAO)/STO, in comparison with the samples grown by PLD (see the open green points), lower R_{sheet} can be realized by using MBE (see the filled green points). This trend is probably due to the suppression of interface roughness scattering. We note that the 2DEG at the γ -Al₂O₃ (GAO)/STO interface has a low R_{sheet} of $\sim 1.5 \text{ } \Omega/\square$ with n_{sheet} of $\sim 8.5 \times 10^{13} \text{ cm}^{-2}$, even though PLD was used for the growth (see the blue open points)⁷. The light orange area in Fig. R10 expresses the range of R_{sheet} and n_{sheet} reported for various STO interfaces. Our data of R_{sheet} of $\sim 2.4\text{--}8 \text{ } \Omega/\square$ with $n_{\text{sheet}} \sim 8.9 \times 10^{13} \text{ cm}^{-2}$ obtained in the LSMO/LTO/STO and LTO/STO heterostructures (see the dark orange region) are well within this range.

Supplementary Fig. 10| R_{sheet} vs. n_{sheet} at various STO interfaces. The red, green, blue and light blue points correspond to the data of LTO/STO, LAO/STO, GAO/STO and AlO_x/STO , respectively. The filled and open points are the data of samples made by MBE and PLD, respectively (only the light blue point corresponds to a sample made by sputtering). The light orange zone is the range of R_{sheet} and n_{sheet} for various STO interfaces reported thus far^{S1-S12}. The dark orange area is the range of R_{sheet} obtained in our LSMO/LTO/STO and LTO/STO heterostructures.

Comment 2:

-I think the discussion of the authors on my comment number 7 on why a poorly metallic interface could be better than a semiconducting interface should appear somewhere in the manuscript. As it might appear to be a key ingredient to optimize the spin to charge conversion efficiency, and could be more straightforwardly compatible with electrical injection of spin current avoiding the high resistance of a tunnel barrier.

Our response:

We added the following explanation about the importance of the poorly metallic layer on STO for obtaining efficient spin-current propagation and a long momentum relaxation time of the 2DEG carriers.

➤ **Corresponding revised parts in the manuscript:**

✧ We added the following discussion in line 216–240 on page 9–10 in the revised

manuscript:

(New) **Discussion**

For obtaining a large IEE efficiency, we need large α_R , large τ_e and efficient spin-current propagation. When the nonmagnetic (NM) interlayer between the ferromagnetic (FM) layer and STO is insulating, such as in FM/LAO/STO^{3,4,34} and FM/AlO_x/STO^{5,6}, most of the converted charge current flows at the interface between the NM layer and STO. In this case, we can use the large τ_e of the 2DEG at this interface. However, the spin current injected from the ferromagnetic layer is attenuated when passing through the insulating NM layer before reaching the NM/STO interface¹⁴⁻¹⁶, leading to a decrease in the IEE efficiency. The large loss for the spin current in diamagnetic insulators has been experimentally demonstrated by spin pumping measurements⁴¹. Meanwhile, when the NM interlayer is purely metallic, the spin current is more efficiently injected into the 2DEG region; however, part of the converted current diffuses to the NM layer, effectively decreasing τ_e and thus the IEE efficiency⁴². Therefore, choosing an appropriate interlayer material that can maximize both τ_e and the spin current propagation is crucial for obtaining efficient IEE.

Our result indicates that LTO is a suitable material from this point of view. The spin current can propagate in LTO very efficiently because of its metallicity. Nevertheless, the resistivity ρ of LTO is not so low; ρ is only about $1 \times 10^{-3} \Omega \cdot \text{cm}^{20}$, which is much higher than that of ordinary metals. This “moderate” feature of LTO can prevent the reduction in τ_e of the 2D electrons. Actually, the LSMO/LTO bilayer has a relatively high R_{sheet} of $\sim 200 \Omega/\square$ (on the LSAT substrate) (Fig. 2a). Meanwhile, R_{sheet} of the 2DEG at the LTO/STO interface is $\sim 2 \Omega/\square$, which is two orders of magnitude smaller than that of the LSMO/LTO bilayer as shown in Fig. 2a. Therefore, in our LSMO/LTO/STO sample, most of the current flows at the LTO/STO interface so that τ_e is not reduced significantly. In addition, the sharpness of the interface in our heterostructures is also thought to substantially increase τ_e due to the suppression of interface roughness scattering.

Reviewer #2's comments

The authors have responded satisfactorily to the comments of the two reviewers. This reviewer recommends the present manuscript for publication in Nature Communication. The reviewer highlights that this new system shows a record spin-charge current conversion efficiency, reaching raw data of 400 μV dc spin pumping voltage for a

microwave power input of 100 mW, and an IEE efficiency value of almost 200 nm. This work sets another precedent for the spintronics and oxide community for instance.

Our response:

We would like to really thank the reviewer for the high evaluation of our work.

Another minor revised point

We revise Supplementary Fig. 8 because we missed to correct this figure in the previous revision.

➤ **Corresponding revised part in the manuscript:**

✧ We modified the momentum relaxation time in Supplementary Fig. 8.